# Metabolite G-Protein Coupled Receptors in Cardio-Metabolic Diseases

**DOI:** 10.3390/cells10123347

**Published:** 2021-11-29

**Authors:** Derek Strassheim, Timothy Sullivan, David C. Irwin, Evgenia Gerasimovskaya, Tim Lahm, Dwight J. Klemm, Edward C. Dempsey, Kurt R. Stenmark, Vijaya Karoor

**Affiliations:** 1Department of Medicine Cardiovascular and Pulmonary Research Laboratory, University of Colorado Denver, Denver, CO 80204, USA; Derek.Strassheim@cuanschutz.edu (D.S.); Timothy.sullivan@cuanschutz.edu (T.S.); David.Irwin@cuanschutz.edu (D.C.I.); Evgenia.Gerasimovskaya@cuanschutz.edu (E.G.); Dwight.klemm@cuanschutz.edu (D.J.K.); Edward.Dempsey@cuanschutz.edu (E.C.D.); Kurt.stenmark@cuanschutz.edu (K.R.S.); 2Division of Pulmonary, Critical Care and Sleep Medicine, National Jewish Health Denver, Denver, CO 80206, USA; LahmT@njhealth.org; 3Rocky Mountain Regional VA Medical Center, Aurora, CO 80045, USA; 4Division of Pulmonary Sciences and Critical Care Medicine, School of Medicine, University of Colorado, Anschutz Medical Campus, Aurora, CO 80045, USA

**Keywords:** metabolic syndrome, metabolites, GPCR, cardiovascular, inflammation

## Abstract

G protein-coupled receptors (GPCRs) have originally been described as a family of receptors activated by hormones, neurotransmitters, and other mediators. However, in recent years GPCRs have shown to bind endogenous metabolites, which serve functions other than as signaling mediators. These receptors respond to fatty acids, mono- and disaccharides, amino acids, or various intermediates and products of metabolism, including ketone bodies, lactate, succinate, or bile acids. Given that many of these metabolic processes are dysregulated under pathological conditions, including diabetes, dyslipidemia, and obesity, receptors of endogenous metabolites have also been recognized as potential drug targets to prevent and/or treat metabolic and cardiovascular diseases. This review describes G protein-coupled receptors activated by endogenous metabolites and summarizes their physiological, pathophysiological, and potential pharmacological roles.

## 1. Introduction

Proteins, fats, and carbohydrates are the primary dietary macronutrients used as energy sources and building blocks by cells, and generate metabolites that have signaling functions [1,2,3]. Metabolites generated include lipid metabolites (free fatty acids, ketone bodies, ceramide, prostanoids, leukotrienes, bile acids), TCA cycle intermediates, (succinate, and α-ketoglutarate amino acids (taurine, phenylalanine, tryptophan), and nucleosides (Adenosine, ATP, UTP, ADP, β-NAD) [4]. Aside from diet, the gut microbiome plays a significant role in generating metabolites, and dysbiosis of the gut can cause an imbalance in metabolites and disease [5,6,7,8].

Metabolic syndrome is a combination of comorbidities, including chronic low-grade inflammation, obesity, elevated blood pressure, impaired glucose tolerance, insulin resistance, and dyslipidemia [9,10]. The development of these comorbidities is a multi-factorial process involving many tissues, including adipose, skeletal muscle, liver, pancreas, and vasculature. Recent studies have identified several G protein-coupled receptors (GPCRs) that bind nutrient metabolites and influence many metabolic processes, including glucose homeostasis and insulin secretion, appetite, calcium-sensing, heart rate, and blood pressure [11,12,13,14]. Therefore, they were recognized as potential drug targets to prevent and treat metabolic and cardiovascular diseases [12,15,16]. This review focuses on GPCRs activated by endogenous metabolites (lipid, Tricarboxylic Acid (TCA) cycle, amino acid, and nucleotide). It summarizes their role in obesity, insulin resistance, hypertension, and inflammation associated with cardiometabolic syndrome.

## 2. Lipid Metabolites

In metabolic diseases, genetic factors, diet, and the gut microbiome are external factors that influence dysregulations leading to obesity, T2D, and dyslipidemia, which contribute to cardiovascular disorders. This section will discuss the role of GPCRs that bind lipid metabolites, which include free fatty acids (FFAs), ketone bodies, carboxylic acids, prostanoids, ceramides, and leukotrienes. Here we describe the role of these GPCRs in cardiometabolic diseases [17,18,19].

### 2.1. Free Fatty Acid Receptors (FFAR)

Fatty acids are carboxylic acids with a long aliphatic chain and are classified based on their chain length as short-chain fatty acids (SCFA)s, containing less than six carbon atoms), medium-chain fatty acids (MCFA)s, 6–12 carbons), and long-chain fatty acids (LCFA)s, 12 or more carbons). The primary source of SCFAs is the bacterial fermentation of dietary fibers or fermented food products [20]. MCFAs and LCFAs are derived mainly from dietary triglycerides. Under physiological conditions, FFAR promotes insulin and incretin hormone secretion, adipocyte differentiation, and anti-inflammatory effects. FFAR_2_ (GPR43), FFAR_3_ (GPR41), and olfr78 bind short-chain fatty acids; FFAR_1_ (GPR40) and FFAR_4_ (GPR120) bind MCFA and LCFA. Thus, these GPCRs act as fatty acid sensors with selectivity for a carbon chain length of FFA derived from food or food-derived metabolites [21].

#### 2.1.1. Short-Chain Fatty Acid Receptors (SCFA)

SCFA are products of the intestinal microbial fermentation of indigestible foods; complex carbohydrates, such as resistant starch or dietary fiber. SCFA are important in gut health and act as signaling molecules in the systemic circulation, affecting the metabolism and peripheral tissue function. GPR41 (FFAR_3_), GPR43 (FFAR_2_) Olfr78, and GPR109A (hydroxycarboxylic acid receptor 2, HCA_2_ receptor) bind SCFAs acetate, propionate, and butyrate with different affinities. Acetate and propionate preferentially activate GPR43, and propionate and butyrate activate GPR41. GPR41 couples to Gi/G0 protein while GPR43 acts via Gq/11 and Gi/G0 proteins. They are expressed in adipose tissue, skeletal muscle liver, leukocytes, vasculature, and the gastrointestinal tract [22].

The binding of SCFAs (propionate, acetate, butyrate) to GPR43 and GPR41 reduces lipolysis, insulin sensitivity, and fat accumulation in white adipose tissues (WAT) by inhibition of hormone-sensitive lipase [22]. Some recent studies show that intake of exogenous acetate, propionate, or butyrate prevents weight gain in diet-induced obesity (DIO) in mice and overweight humans. SCFAs alter gut microbiota compositions, increase the release of the anorectic gut hormones PYY and GLP-1 from enteroendocrine cells, reducing food intake and body weight gain [23]. In addition, SCFAs prevent beige adipogenesis and mitochondrial biogenesis in the adipose tissue, resulting in enhanced triglyceride hydrolysis and FFA oxidation and increased leptin secretion. The fecal SCFA concentration is increased in genetically obese (ob/ob) mice and obese humans and is speculated to be due to a decrease in transporters required for the absorption of SCFA. In a human cohort, SCFA levels in plasma were inversely proportional to blood pressure, while fecal SCFA content was positively associated with blood pressure [24]. Mice on HFD had lower fecal levels of SCFA, suggesting an increase in absorption, and conversely, germ-free mice with low gut SCFA levels were protected from diet-induced obesity (DIO) [25,26].

Studies show that GPR43^−/−^ mice are obese on a regular diet but have a lower body fat mass and increased lean mass on HFD with improved insulin sensitivity than mice on a regular diet [27]. GPR43 deficiency decreased macrophage infiltration to WAT and increased the activity of brown adipose tissue (BAT), suggesting an increase in energy expenditure leading to weight loss. Paradoxically mice overexpressing GPR43, specifically in adipose tissue, are also protected against diet-induced obesity by suppressing insulin signaling and increasing the consumption of lipids [28]. Therefore, understanding the tissue-specific roles of GPR43 may be essential to determine its exact role in obesity. GPR43 levels in β-Cell and serum levels of acetate are increased with HFD [29]. Thus, GPR43 KO mice on HFD develop glucose intolerance due to a defect in insulin secretion. However, there are some inconsistencies on the effect of GPR43 KO on glucose levels from different studies, which need to be resolved before therapeutic use in targeting diabetes [30].

Gender differences were observed in GPR41 KO on HFD, with only male mice exhibiting a higher body fat mass than wild-type littermates. GPR43 agonists induced the differentiation of mouse but not human adipocytes [31]. Studies using primary cultured adipocytes derived from GPR43 or GPR41 KO mice demonstrated that GPR43 mainly regulates SCFA-induced the suppression of lipolysis and the secretion of leptin [32].

GPR41 modulates insulin secretion and gene expression in pancreatic β-cells and modifies metabolic homeostasis in fed and fasting states [33]. Transgenic mice overexpressing GPR41 showed decreased glucose responsiveness [34]. GPR41 KO mice showed fasting hypoglycemia, consistent with increased basal and glucose-induced insulin secretion by islets in vitro [34].

SCFAs suppress atherosclerotic lesions and inflammation in ApoE^−/−^ mice [35,36]. In contrast, another study showed that eliminating the microbiota in ApoE^−/−^ deficient mice caused a significant reduction in atherosclerotic lesion formation. In addition, these mice had a significant increase in plasma and hepatic cholesterol concentrations, suggesting that the beneficial effects were due to attenuation of inflammatory responses [37].

The heart depends primarily on glycolysis and lactate oxidation to produce energy in the embryonic stage and shifts to utilizing fatty acids after birth [38]. In failing hearts, the metabolism shifts more towards glycolysis [39]. SCFAs binding to GPR41 and OlfR78 had opposing effects on blood pressure (BP). Oral administration of SCFAs stimulated GPR41 and decreased BP, whereas stimulation of Olfr78 raised BP [40]. GPR41 and GPR43/109A KO mice have a significantly larger heart-to-body weight index, higher end-diastolic and pulse pressure, and perivascular fibrosis than wild-type mice.

In contrast, Olfr78-deficient mice displayed lower renin concentrations and decreased BP [32,41]. Reducing the microbiota by antibiotic treatment in OlfR78-knockout mice reduces SCFAs in the gut and increases BP due to lack of ligand to bind GPR41 and promote hypotension [41,42]. GPR41 is expressed in endothelial cells in the vasculature and OlfR48 in smooth muscle cells [41]. Propionate administration decreased blood pressures by decreasing active vascular tone [43] The hypotensive effect of propionate was not observed in *GPR41*-deficient mice [43].

GPR41 and GPR43 are expressed in polymorphonuclear leukocytes and phagocytes, couple to Gi/Gαq, and mediate chemotaxis-phagocytosis-respiratory burst [44]. Studies suggest that GPR41 and GPR43 may exert both pro-and anti-inflammatory effects, depending on the disease model used. The anti-inflammatory effects of SCFA effects on HDACs and NF_K_B mediate anti-inflammatory responses [45]. SCFA receptor GPR43 regulates inflammatory signals by modulating macrophage phenotype in adipose tissues. GPR41 protects against mechanical-wire mediated arterial injury, a process that involves the mac–neutrophil axis. Supplementation with propionate promotes the anti-inflammatory response of Treg cells to reduce local infiltration of immune cells, thereby reducing cardiac hypertrophy and fibrosis, susceptibility to cardiac arrhythmias, and atherosclerotic lesion burden and exhibits antihypertensive effects in angiotensin II (Ang II)-induced hypertension or atherosclerosis [46]. In db/db mice, butyrate suppresses obesity-induced inflammation in adipose tissues by inhibiting the NOD-like receptor 3 (NLRP3) inflammation signaling pathway [47]. In explants of human omental and subcutaneous adipose tissues, propionate suppresses expression of the adipocyte-derived proinflammatory cytokine, resistin [33].

GPR41 and GPR43 have a more conformed role in fat metabolism. Olfr78 expression is associated with blood pressure. Although studies have indicated a causal role for SCFAs in metabolic health, the effects are variable [48]. The inconsistent knockout phenotypes observed in different studies need to be addressed [49]. Since similar effects observed with KO and overexpressor mice also warrant further studies that may include tissue-specific effects of the different receptors and correlating outcomes with gut microbiota composition and metabolism. In addition, the species-specific effects observed in adipocytes questions the translatability of mice to human. Therefore, future studies on human-derived cells and tissues are required to understand their role in metabolism. Finally, factors such as diet and physical activity could also influence outcomes and need to be assessed.

#### 2.1.2. Medium and Long-Chain Fatty Acid Receptors

MCFA and LCFA are derived from dietary fat intake or metabolic turnover of triglycerides. In humans. MCFAs and LCFAs are metabolized by β-oxidation and utilized as an energy source in various tissues [20]. MCFAs reduce adiposity in obese individuals while LCFAs increase adiposity [50,51]. GPR84 binds MCFA, and GPR40 GPR120, GPR119 binds LCFA.

GPR84: GPR84 binds MCFAs, and is expressed in immune cells from bone marrow, spleen, lung, lymph nodes, and adipose tissue [52,53]. GPR84 is predominantly a proinflammatory receptor and links fatty acid metabolism and immune responses; however, studies are limited [54]. GPR84 mRNA is increased in fat pads of mice on HFD [55]. However, deletion of GPR84 did not affect body weight or glucose tolerance in mice fed either a high MCFA or LCFA diet. GPR84 levels are increased by inflammatory cytokines such as TNF-α and IL-1 in human adipocytes [55,56,57]. Similar observations were made in mouse 3T3-L1 adipocytes with TNF-α and LPS treatment and in human adipose-derived stem cells [58]. More studies are required to understand how these inflammatory signals increase GPR84 and its role in metabolism. GPR84 KO mice show an increase in liver triglycerides on the MCFA diet and myocardial triglycerides on LCFA diets. GPR84 expression was also increased in livers of patients with nonalcoholic fatty liver disease (NAFLD) [54].

An increase in GPR84 is seen in diabetes, atherosclerosis, and other diseases associated with inflammation [56,59]. GPR84^−/−^ mice on MCFA-enriched diet exhibit glucose intolerance and a defect in insulin secretion, which was not reproduced in a different study [60]. MCFA-fed KO mice also exhibit mitochondrial dysfunction in the skeletal muscle paradoxically with increases in mitochondrial content [61]. High glucose concentrations, oxidized LDL (oxLDL), and LPS increased GPR84 expression in macrophages [62].

*GPR84* mRNA levels are higher in ApoE^−/−^ mice on HFD. GPR84 agonists also increase cholesterol efflux and are reported to be protective in atherosclerosis [63]. In addition, GPR84 receptor agonists increase inflammatory mediator levels, bacterial adhesion, and phagocytosis in macrophages [64].

A recent study found that GPR84 is upregulated in the lungs of rats with heart failure after myocardial infarction and may have a role in the progression of lung fibrosis [65]. GPR84 inhibitors significantly reduced markers of inflammation and fibrosis and are in clinical trials for the treatment of IPF [54]. In addition, transforming growth factor-beta (TGFβ) and endothelin 1 increase GPR84 expression in cultured human lung fibroblasts.

Available studies on GPR84 in cardiometabolic syndrome show that it has a proinflammatory role in the processes of diabetes and atherosclerosis [66]. MCFA may activate macrophages via the GPR84 receptor. However, future studies using tissue-specific KO will be required to understand its physiological role in different tissues. For instance, macrophage-specific GPR84 KO mice in HFD and diabetic models will clarify whether its expression in macrophages contributes to inflammation. In addition, a better understanding of its protective role in atherosclerosis and whether effects on the heart and lung are solely due to inflammation or whether it has metabolic effects [58].

GPR40/FFAR1 Receptor. GPR40/FFAR1 is activated by LCFAs, mainly oleic acid, and is expressed in pancreatic β cells, intestinal cells, immune cells, splenocytes, and the brain [67]. The activation of GPR40 is linked primarily to the modulation of the Gq family G proteins and intracellular calcium. Activation of Gs- and Gi-proteins to modulate intracellular levels of cAMP were also reported [68].

GPR40 protein levels are increased in the pancreas of Zucker fa/fa rats [69]. GPR40 KO mice are protected from obesity-induced hyperinsulinemia, hepatic steatosis, and impaired glucose tolerance, whereas chronic overexpression in β-cell causes hypo-insulinemia and diabetes [70]. A subsequent study found that GPR40-deficient mice are hyperglycemic on fasting and not protected from HFD-induced insulin resistance and liver steatosis [70,71]. Nevertheless, another study shows that GPR40 contributes to the maintenance of basal metabolism, and GPR40^−/−^ mice had increased body weight, higher insulin levels, insulin resistance, cholesterol, FFA on an LFD [72,73,74]. These studies suggest that GPR40 may have a homeostasis role in metabolism and may not contribute to pathology.

The interaction of lipids and glucose on the regulation of GPR40 protein levels and hormone secretion in pancreatic endocrine cells is essential in the pathogenesis of obesity and T2D [75]. FFAs increased GPR40 expression, while high glucose decreased GPR40 protein expression [76]. FFA-induced release of islet hormones in Goto-Kakizaki (GK) rats that are non-obese hyperglycemic and in fa/fa rats that are mildly hyperlipidemic obese but normoglycemic is dependent on GPR40 protein expression [75]. MR1704, a GPR40 agonist, improved glucose homeostasis through glucose-dependent insulin secretion (GSIS) with a low risk of hypoglycemia and pancreatic toxicity in the GK rats. Chronic activation of GPR40 in transgenic mice overexpressing GPR40 in pancreatic β-cells augmented glucose-stimulated insulin secretion and improved glucose tolerance [77]. SiRNA- or oligonucleotide-mediated reduction of GPR40 expression in β-cell lines or isolated mouse pancreatic islets reduces augmentation of insulin secretion by FFAs. GPR40 antagonists, such as GW1100, inhibit GPR40-mediated augmentation of insulin secretion from MIN6 cells.

GPR40 helps in the secretion of several incretins such as cholecystokinin, glucagon-like peptide-1 (GLP-1), the gastric inhibitory peptide (GIP), peptide YY (PYY) [78]. The beneficial anti-diabetic and anti-inflammatory effects of palmitic acid, hydroxy stearic acids are dependent on the expression of GPR40 [79]. GPR40 reduces insulin secretion in response to fatty acids in vivo and in vitro without affecting the response to glucose [71].

GPR40 agonists might be effective insulin secretagogues for treating type 2 diabetes. GPR40 agonists were used for the treatment of diabetes in clinical trials but have shown conflicting results. Drugs targeting GPR40 have failed in clinical trials due to hepatic toxicity. Future studies addressing the function of GPR40 on other insulin-sensitive tissues such as adipose, liver, and skeletal muscle will help to understand its role in T2D better.

GPR120/FFAR4 The GPR120 receptor is a Gαq-coupled GPCR expressed in many tissues, including the liver, adipose tissue, intestines, macrophages, and pancreas. It binds alpha-linolenic acid (ALA), eicosapentaenoic acid (EPA) palmitate, myristic acid, and oleic acid (OA) and docosahexaenoic acid (DHA) [64]. Genetic mutations of GPR120 in both humans and mice are linked to obesity, increased fasting glucose levels, and insulin [80]. GPR120 expression increases in white adipose, cardiac, and skeletal muscle tissues of mice or rats on a high-fat diet [81]. GPR120 activation relieves insulin resistance by enabling adipogenesis in adipose tissue and adipocytes and inhibiting lipolysis [80,82]. GPR120-deficient mice on HFD had decreased expression of Insulin signaling-related genes in adipose tissue and the liver of HFD-fed [81,83]. GPR120 KO leads to impaired adipocyte differentiation, enhanced insulin resistance, and glucose intolerance with HFD [84]. In T1D/T2D protective mechanisms, GPR120 stimulated brown adipose tissue to generate heat, increasing FAO-UCP [85]. Activation of GPR120 in human pancreatic islets using eicosapentaenoic acid decreased lipid-induced apoptosis and protected pancreatic islets from lipotoxicity [86,87]. In addition, it increases insulin sensitivity by increasing the incretins GLP-1 in pancreatic β cells and the gut fatty acid-induced secretion of cholecystokinin (CCK) [88].

GPR120 KO mice are unable to adapt to pressure overload induced by transverse aortic constriction [89,90]. GPR120 stimulates ABCA_1-_ABCG_1_-mediated cholesterol efflux and is protective against atherosclerosis [91]. In humans, GPR120 expression is decreased in heart failure [92], while the R270H polymorphism correlated with an eccentric remodeling in a large clinical cohort [90]. GPR120 agonists protect endothelial cells from oxLDL induced toxicity by decreasing E-selectin/VCAM_1_ expression [93]. GPR40 and GPR120 are expressed on airway smooth muscle and modulate airway smooth muscle tone and may have a role in obesity-induced asthma [94].

GPR120 expression is increased in macrophages in adipose and liver of obese mice [95]. Activation of GPR120 by ω-3 FFAs has insulin-sensitizing and anti-diabetic effects in vivo due to the repression of macrophage-induced tissue inflammation [96]. Defective macrophage efferocytosis in ob/ob macrophages can be reversed by treatment with EPA or by feeding ob/ob mice a ω3-rich diet, demonstrating the beneficial effects of ω3 supplements in genetic models of obesity [97].

GPR120 agonism with ω3 FA supplementation may be helpful in the prevention of metabolic disorders such as obesity and diabetes. In addition, GPR120 agonists with improved selectivity were developed. Given its role in controlling inflammation, targeting this receptor could have therapeutic potential in many inflammatory diseases, including obesity and T2D, and cardiovascular disease.

GPR119: GPR119 is expressed in β-cells in the pancreas, gastrointestinal tract, and fetal liver. Low levels in cardiac and skeletal muscle in humans were also reported [98]. GPR119 is activated by oleic acid-containing lipids such as oleoyl ethanolamide (OEA) and maintains glucose homeostasis by releasing GLP-1 from L-cells and insulin from β-cells [99]. GPR119^−/−^ mice had defects in GLP-1 release but were not found to differ significantly from wild-type littermates in size, body weight, or blood glucose levels in the fed or fasted state [100]. In various animal models of obesity and T2D, synthetic GPR119 agonists lowered blood glucose without hypoglycemia, slowed diabetes progression, and reduced food intake and body weight [101,102]. Glucose tolerance and insulin responsiveness to the glycemic challenge were not impaired in β cell-specific GPR119 knockout mice exposed to standard chow or high-fat diets [103]. Recent studies suggest the role of GPR119 as a therapeutic target for the hypophagic action of OEA, is independent of the receptor [104]. GPR119 agonists may have a promising role in the treatment of T2D and related metabolic disorders. However, clinical trials with agonists of GPR119 were disappointing [105].

GPR119 decreases metabolism in cardiac and skeletal myoblasts and lipid status can influence signaling pathways [106]. Lentiviral expression of GPR119 reduced cholesterol levels by inhibiting Ox-LDL uptake and enhancing cholesterol efflux in THP-1 macrophage-derived foam cells. Infection of ApoE^−/−^ mice with lentiviral GPR119 reduced serum levels of lipids and inflammatory cytokines and prevented plaque formation [101]. A clinical study in dyslipidemia patients demonstrated that administration of GSK1292263 enhanced plasma HDL-cholesterol levels and significantly reduced LDL-cholesterol and triglyceride levels compared to placebo [107]. The detrimental effect of the decrease in oxidative/metabolic capacity warrants more studies before GPR119 agonists can treat metabolic diseases [104,108,109,110].

### 2.2. Ketone Bodies

Ketone bodies are endogenous metabolites produced by the degradation of fatty acids via β-oxidation to form acetyl-CoA in the liver during fasting, insulin deprivation, and exercise [111]. When carbohydrates are low, ketone bodies are used as an energy source in the brain, heart, and skeletal muscle [112]. The metabolism of ketone bodies interfaces with multiple processes, including the tricarboxylic acid cycle, β-oxidation of fatty acids, de novo lipogenesis, sterol biosynthesis, glucose metabolism, the mitochondrial electron transport chain, and hormonal signaling [113]. They are important as signaling mediators and promote post-translational modification of proteins, inflammation, and oxidative stress [114]. Ketone bodies are increased in T1D and T2D, and heart failure, and during aging [115]. Growing evidence suggests that ketones may be beneficial for patients with cardiovascular disease. The GPCRs HCA_1_/GPR81, HCA_2_/GPR109A, and HCA_3_/GPR109B are receptors for the ketone bodies (acetoacetate, lactate, 3-hydroxybutyrate, and β-hydroxy octanoate). These hydroxy-carboxylic acids (HCAs) serve as intermediates of energy metabolism and protect against the pathological effects of ketone bodies-ketoacidosis under changing metabolic and dietary conditions [116].

#### Hydroxycarboxylic Acid Receptors (HCA)

HCA receptors are expressed in adipose tissue and mediate anti-lipolytic effects through G_i_-dependent inhibition of adenylyl cyclase and decrease serum fatty acids, thereby reducing serum fatty acid, thereby liver generated ketone bodies [117]. Three subtypes of HCA receptors were identified and bind to different endogenous metabolites, regulate lipolysis in a negative feedback manner, and thus function as metabolic sensors [118]. The HCA_1_ receptor is activated by the glycolytic metabolite 2-hydroxy-propionic acid (lactate), the HCA_2_ receptor is activated by the ketone body 3-hydroxy-butyric acid HCA_3_ receptor is by the β-oxidation intermediate 3-hydroxy-octanoic acid [117]. While HCA_1_ and HCA_2_ receptors are present in most mammalian species, the HCA_3_ receptor is exclusively found in humans and higher primates. HCA_1_ mediates the anti-lipolytic effects of insulin in the fed state [119]. HCA_2_ and HCA_3_ inhibit lipolysis during conditions of prolonged fasting when there is increased β-oxidation [120]. Chronic obesity reduces HCA_1,_ HCA_2_ expression in WAT, but acute inflammation increases HCA_2_ in adipocytes and macrophages [12]. Under metabolic stress, such as diabetes, ketone bodies can dramatically increase serum, raising serum pH ketoacidosis, dangerous to cardiac function [121]. HCA_2_ is a receptor for the anti-dyslipidemia drug nicotinic acid, HCA_1_ and HCA_3_ also represent promising drug targets, and several synthetic ligands for HCA receptors were developed [117].

HCA_1_(GPR81): GPR81 is expressed in adipose tissue with low kidney, skeletal muscle, and liver levels [122]. GPR81 is also localized in the mitochondria [123,124]. GPR81 expression was decreased in obese mice and adipocytes during differentiation [116,122,125,126]. Rosiglitazone, the peroxisome proliferator-activated receptor-γ (PPARγ), increases the transcription of the GPR81 gene by binding to the promoter [127].

Monocarboxylate transporters (MCT)s prevent intracellular accumulation of lactate by removing excess lactate produced due to increased glycolytic activity [128]. Lactate is used as fuel by muscle, and its levels are increased in white adipose tissue in obesity and during exercise [129]. Activation of GPR81 by lactate inhibits lipolysis [123]. GPR81 regulates the ability of white and brown adipocytes to produce heat [130]. GPR81 mRNA expression is upregulated during preadipocyte differentiating into mature adipocytes. Activation of GPR81 lowers blood glucose through increased glucose uptake by BAT in male mice with DIO [131]. GPR81 agonists suppressed fasting plasma FFA levels in rodents and improved insulin sensitivity in mouse models of insulin resistance and diabetes [122]. GPR81 KO mice exhibited insulin-induced increased lipolysis in white adipose tissue and significantly attenuated experimental adipose browning and thermogenesis [116,120].

Agonists for GPR81 induced hypertension in wild-type, but not GPR81-deficient mice [122]. In dogs, the pressor effects were shown to be due to increased vascular resistance. GPR81 agonism in blood pressure control and regulation of renal vascular resistance by modulation of the endothelin system [122,132]. TNF-α and IL-1β, secreted by macrophages, increase GPR81 expression [133,134]. Selective activation of GPR81 may serve as a novel treatment target against endothelial inflammation and the development of atherosclerosis induced by oscillatory shear stress [135,136]. Blood lactate is associated with carotid atherosclerosis and is related to insulin resistance [137].

Although GPR81 is expressed in macrophages and dendritic cells, its role in inflammation is not well studied [133]. It exhibits anti-inflammatory effects by inhibiting inflammasome formation and activation of NFkB by a mechanism that involves β-arrestin2 [138]. GPR81 inhibits inflammatory signaling pathways in macrophages of the liver, spleen, and pancreas. However, in other studies, GPR81 independent effects of lactate on inflammation were also reported [139].

Further in vivo studies are required for GPR81 antagonists to be made helpful in treating diet-induced obesity [116]. Furthermore, the tissue-specific effects of the receptor may be of concern, and identification of other endogenous ligands is also essential. In addition, the cross talk of lactate with GPR109A, at least in cancer, was predicted to have opposite effects, and this interaction needs further characterization.

HCA_2_ (GPR109A) GPR109A is a Gi-coupled receptor expressed predominantly on adipocytes and immune cells. GPR109A is a receptor for nicotinic acid, and later studies showed that the receptor is activated by the ketone body 3-hydroxy-butyric acid(β-OHB) [117]. β-OHB is synthesized in the liver from FFAs or derived from lipolysis in adipocytes and inhibits lipolysis during starvation [140]. Thus, GPR109A modulates de novo lipid accumulation in liver and adipose tissue, and its dysregulation can lead to age-associated obesity and hepatic steatosis. GPR109A is responsible for niacin-mediated inhibition of lipolysis and increased secretion of adiponectin. GPR109A agonists that modulate lipid and adiponectin concentrations are being tested in clinical trials [141]. Niacin does not decrease plasma FFA or TG levels In GPR109A^−/−^mice [142]. Mice on HFD have decreased expression of GPR109A in adipocytes and a decrease in basal and catecholamine-induced lipolysis [126].

In contrast, GPR109A levels were increased with LPS treatment in 3T3L1 adipocytes and Raw macrophages, suggesting a potential role in the crosstalk between metabolic and inflammatory pathways [126,143]. GPR109A decreases inflammation in adipose tissue because LCFA released by WAT is a major promoter of vascular inflammation [144]. Activation of GPR109A by butyrate in macrophages decreases activation of the NLRP3 inflammasome, NFkB activation by decreasing phospho-p65, the induction of TNFα, IL6, IL1, and M1 phenotype [145,146,147,148,149,150,151]. In particular, studies report that niacin can reduce inflammation in atherosclerosis, obesity, sepsis, diabetic retinopathy, and renal disease [152]. GPR109A expression is increased in macrophages treated with interferon γ [153]. In addition, GPR109A activation promotes neutrophil apoptosis and inhibits myeloperoxidase (MPO) release, thereby suppressing oxidative stress [154].

At pharmacological doses, nicotinic acid reduces plasma concentrations of VLDL and LDL cholesterol, triglycerides, and lipoprotein while increasing HDL cholesterol levels [150,155,156]. In Ldlr^−/−^ mice, niacin protected against the progression of atherosclerosis. The vascular protective effects of niacin in atherosclerosis are abolished in mice with deletion of GPR109A in bone marrow-derived cells and Ldlr^−/−^ GPR109A^−/−^ mice [157,158]. GPR109A activation by β-OHB can cause vasodilation of isolated resistance arteries [159,160]. Niacin attenuated the development of hypoxia/SU5416–induced PH in mice and suppressed the progression of monocrotaline-induced and hypoxia/SU5416–induced PH in rats through reducing pulmonary artery remodeling [161,162]. Niacin protects against aortic aneurysms independent of GPR109A, most likely by serving as an NAD^+^ precursor [157]. This cardioprotection by prebiotic fiber effect involves SCFA receptors, especially GPR43 and GPR109A [163]. While HCA2 is an established target for drugs such as nicotinic acid, which have anti-dyslipidemia and anti-atherogenic effects, activation of GPR109A may have additional anti-inflammatory and immunomodulatory effects that have not been explored yet but warrant further investigation [150].

Currently, clinical studies evaluate the efficacy of nicotinic acid in combination with statins in reducing relevant clinical endpoints, including progression of cardiovascular disease, the incidence of major cardiovascular events, and associated mortality. It was observed that SNPs in GPR109A determine response to therapy by niacin in the lowering of lipoproteins [164]. A recent epidemiological analysis of adverse reactions with niacin indicated severe GI symptoms and anxiety. Since GPR109A has beneficial metabolic and anti-inflammatory effects, a better understanding of the mechanisms of both the desired and adverse effects will allow for broader use of the drug [165].

HCA_3_ receptor (GPR109B) GPR109B is a receptor for the β-oxidation intermediate 3-hydroxy-octanoic acid and is expressed exclusively in humans and higher primates [100,117]. HCA_3_ is highly expressed in human white adipose tissue [116,166]. PPARγ agonists induced expression of HCA_3_ in human multipotent adipose-derived stem cells [117]. HCA_3_ inhibits lipolysis during physiological and pathophysiological conditions of increased β-oxidation and ketogenesis, preventing keto-acidosis and promoting the efficient use of fat stores [167]. HCAR_3_ is activated by kynurenic acid, an intermediate in the kynurenine pathway of tryptophan degradation that is an agonist of another GPCR, GPR35 [168]. HCAR_3_ in immune cells such as neutrophils and macrophages has raised questions about the primary functions of HCAR_3_ and its potential as an immunological drug target [169].

HCA_1_ and HCA_3_ also represent promising drug targets, and several synthetic ligands for HCA receptors were developed [170,171]. GPR109B in humans inhibits lipolysis under conditions of physiological or pathological increases in β-oxidation rates. HCA_3_ was expressed in various human immune cells and activated by endogenous agonists resulting in intracellular calcium release [117]. HCA_2_ and HCA_3_ ligands modulated LPS-mediated proinflammatory gene expression in both human macrophages and adipocytes without affecting adipogenesis [12,32]. Therefore, targeting HCA_2_ and HCA_3_ would be beneficial in treating inflammation conditions associated with atherosclerosis and obesity-related adipose tissue inflammation. However, an understanding of the role of HCA_3_ is lacking due to animal models expressing the human receptor. Humanized mice models expressing HCA_3_ will help in understanding its function in vivo. Several HCA_3_-specific agonists were synthesized that are expected to inhibit lipolysis in human adipocytes [172]. Future work is required to understand the function of HCA3 in humans and to explore whether it is of use as a drug target with advantages compared to HCA2.

### 2.3. Bile Acids

TGR5/Gpbar1 G-protein-coupled bile acid receptor1 (Gpbar1)/TGR5 is a GPCR that binds bile acids generated by cholesterol catabolism and effects, bile acid homeostasis, energy homeostasis as well as insulin signaling, and inflammation [173,174,175,176,177]. TGR5 mRNA is detected in the small intestine, stomach, liver, lung, and spleen.

Dysfunctions in bile acid metabolism cause cholestatic liver diseases, dyslipidemia, fatty liver diseases, cardiovascular diseases, and diabetes [175]. TGR5 receptor, expressed in adipocytes, regulates energy expenditure and body weight [177]. Bile acids increase oxygen consumption and extracellular acidification rate in BAT and human skeletal muscle cells [178]. TGR5^−/−^ mice have decreased bile acids and accumulate fats when fed a high-fat diet [179,180]. Different studies have shown Increased circulating bile acid levels in obese individuals and positively correlated with body mass index [181,182,183].

Bile acids improve glycemic control by activation of TGR5 and increase GLP-1 secretion [184,185,186,187]. Single nucleotide polymorphisms (SNP) at the TGR5 locus (rs3731859) are associated with BMI, intramyocellular lipids, and fasting GLP-1 levels. Changes in bile acid composition were verified in clinical trials and animal models of T2D. Increased concentrations of deoxycholic acid (DA) and decreased concentrations of chenodeoxycholic acid (CDCA) were observed in T2DM patients. TGR5 activation inhibits kidney disease in obesity and diabetes by inducing mitochondrial biogenesis. Agonist for TGR5 improves glucose tolerance, decreases fasting blood glucose and the glycosylated hemoglobin A1c in T2D mice [188]. It was postulated that TGR5 activation in macrophages may prevent insulin resistance and treat T2D [189].

TGR5 signaling may play a critical role in protection against inflammatory diseases, including fatty liver disease, inflammatory bowel diseases, atherosclerosis, and diabetes [190]. Hydrophobic bile acids are considered to be pro-inflammatory, whereas hydrophilic bile acids are anti-inflammatory [191]. Treatment of ApoE- and LDL receptor- knockout mice fed a Western-type diet supplemented with TGR5 agonist reduced atherosclerotic plaque formation and decreased levels of circulating proinflammatory cytokines and chemokines in aortic tissue [192,193].

Bile acid-activated FXR and TGR5 a GPCR suppress inflammation in macrophages, intestine, and hepatocytes by inhibiting NF-κB nuclear translocation and antagonizing NF-κB-dependent induction of induction proinflammatory cytokines [174,192,194,195]. Activation of TGR5 by INT-777 treatment in macrophages inhibited cytokine production through cAMP-NF-κB. Bile acid-induced GLP1 also exerts beneficial effects on endothelial function, blood pressure, myocardial metabolism, left ventricular ejection fraction, atherosclerosis, and response to oxidative injury induced by ischemia-reperfusion [196]. Bile acids also activate other GPCRs, sphingosine-1-phosphate receptor 2 (S1PR2) and muscarinic receptor 2 [197]. Conjugated bile acids activate S1PR2 to regulate inflammation in some liver diseases [198,199,200]. Bile acids, bile acid derivatives, and bile acid sequestrants are therapeutic agents for treating chronic liver diseases, obesity, and diabetes in humans [191].

Gut microbiota has a role in modulating bile acid pool size, composition, and enterohepatic recirculation. Therefore, it may be essential to correlate microbiota composition to inter-individual differences in bile acid composition and their effects on metabolic risk. TGR5 agonists are promising drugs for treating metabolic disorders such as type II diabetes, obesity, atherosclerosis, and steatohepatitis. Thus, targeting bile acid receptors signaling seems to derive a promising approach for treating metabolic diseases. However, additional detailed pre-clinical research is required to confirm the efficacy of bile acids and bile acid derivatives in such conditions [191].

### 2.4. Ceramide

Ceramide is generated by sequential degradation of plasma membrane and lipoproteins in the lysosome by acid hydrolase [201]. Sphingosine, produced by the degradation of ceramide, can be recycled in the salvage pathway to ceramide or phosphorylated by sphingosine kinases (SphK)s to form Sphingosine 1-phosphate (S1P) [202]. S1P can also be exported out of cells by specific transporters to activate GPCRs. An increase in serum and tissue levels of ceramides was correlated with obesity and insulin resistance. Sub-cellular localization of ceramides in the mitochondria, ER, and nucleus were inversely correlated with insulin signaling, while lipids in the cytosolic fraction showed no relationship [203]. Therefore, an essential function of SphKs in metabolic disease is to remove excess ceramide [204].

S1PR: S1P signals through five specific G-coupled S1P receptors (S1PR) designated S1PR 1–5, and each subtype exhibits differential coupling efficacy to Gα subunits [205,206]. S1PR1-3 are ubiquitously expressed, whereas S1PR4 is predominantly expressed in the immune system and S1PR5 in the central nervous system. S1P formed in the nucleus inhibits HDAC1/2 inhibitor and is involved in the upregulation of enzymes required for lipid metabolism [207]. S1P levels are associated with obesity, insulin resistance, hyperglycemia, dyslipidemia, and hypertension [208]. Plasma S1P levels were elevated in HFD-induced and *ob*/*ob* mice along with obese humans [209]. The SphK1 level was also elevated in obese, type 2 diabetic humans and in hepatic insulin resistance. Elevated S1P in *ob*/*ob* mice, increased cytokine expression in adipocytes [210]. In 3T3-L1 preadipocytes, S1P significantly decreased lipid accumulation in a dose-dependent manner with the downregulation of the transcriptional levels of the CCAAT/enhancer-binding proteins, triglyceride lipase (ATGL), and perilipin, indicating that FTY720 prevented obesity by modulating adipogenesis and lipolysis [211,212].

SphK1 and SphK2, the isoforms of SphK, exert opposite effects in protecting β-cells from lipotoxicity [213]. SphK2 is the metabolically protective factor, whereas the effects of SphK1 are controversial. Although SphK1 and SphK2 catalyze the same reaction, SphK1 inhibition or KO decreases blood S1P, while SphK2 inhibition increases blood S1P. SphK1 and SphK2 were found essential for GSIS in pancreatic β-cells; however, which of the two isoforms is predominant is not known. SphK1(^−/−^) mice developed diabetes and had reduced insulin levels compared with the WT mice. HFD increased pancreatic β-cell mass by 140% in WT mice and decreased to 50% in SphK1(^−/−^) mice. In primary islets isolated from SphK1(^−/−^), mice exhibited higher susceptibility to lipotoxicity, which was eliminated by S1P treatment. In muscle insulin resistance, the role of SphK needs further clarification. In white adipose tissue, SphK1 prevents obesity-associated diabetes, whereas the adipose-specific role of SphK2 is not known.

Recent studies indicate that the ceramide to sphingolipid ratio is essential in regulating insulin action in metabolic disease. Glucose-activated SphK2/S1P is necessary for glucose-stimulated insulin secretion (GSIS) in pancreatic β cells. SphK1 transgenic mice fed an HFD showed increased SphK1 activity in skeletal muscle and insulin resistance. SphK1(^−/−^) mice showed enhanced insulin signaling in adipose and muscle, improved systemic insulin sensitivity, and glucose tolerance [214]. Glucose elevates intracellular S1P by activating SphK2 in MIN6 cells and mouse pancreatic islets [215]. Manipulating S1P levels correlates with GSIS [216]. Decreasing S1P by the knockdown of SphK2 in MIN6 cells or primary islets results in decreased GSIS, whereas the knockdown of the S1P phosphatase, SPP1, leads to a rise in GSIS [216].

A significant association between S1P and TNF-α was observed in overweight adolescents [216]. S1P/S1PR_2/3_ plays a crucial role in regulating M1 type polarization of BMMs and acts by activating the G(α)_i/o_/PI3K/JNK signaling pathway, with potential implications for new approaches to inflammatory liver disease therapy [217]. Our current study provided strong evidence that the S1P–S1PR axis also is involved in sustaining the inflammatory response and the potential therapeutic effect of blocking this axis at the peak of the inflammatory response by inducing a pro-resolution response.

### 2.5. Arachidonic Acid

Arachidonic acid (AA) is an essential ω-6 polyunsaturated fatty acid (PUFA) obtained from poultry, animal meat, fish, seafood, and eggs. Cyclooxygenases (COX) act on AA to generate prostaglandins and thromboxane, lipoxygenases generate leukotrienes, and cytochrome p450 enzymes generate epoxyeicosatrienoic acids [218]. Prostanoids are a subclass of eicosanoids and compose a group of lipid mediators derived from membrane phospholipids by the action of PLA2. Cyclooxygenase and lipoxygenase metabolize the ω-3 PUFA eicosapentaenoic acid to generate anti-inflammatory mediators with different biological actions than those derived from AA [219]. An increase in the omega-6/omega-3 ratio by increased intake of omega-6 PUFAs contributes to thrombosis and proinflammation, leading to a high prevalence of atherosclerosis, obesity, and diabetes, features of metabolic syndrome [220]. COX-1 and COX-2 metabolize AA to PGH, the common substrate for synthesizing prostacyclin PGI_2_, PGE_2,_ and thromboxaneTXA_2_. In addition, COX-2 is a primary source of proinflammatory PGE_2_ and PGI_2_ [221]. COX2 inhibitors increased the risk of adverse cardiovascular events, including myocardial infarction, stroke, systemic and pulmonary hypertension, thrombosis, suggesting a homeostatic role [222].

Arachidonic acid is converted to prostaglandins, PGI_2_, PGE_2_, TxA_2,_ PGF_2α_, and PGD_2,_ ligands for specific GPCRs, including IP Receptor, PGE_2_ receptors (EP_1–4_), TP receptor, FP receptor, PGD receptors (DP_1_ and DP_2_), respectively [223]. Of these receptors, IP, EP2, EP4, and DP1 are involved in vasorelaxation, and EP1, EP3, FP, and TP promote vasoconstriction [224]. Furthermore, EP_2_, EP_4_, IP, and DP_1_ receptors activate adenylyl cyclase via G_s_, increasing intracellular cAMP. In addition, EP_1_, FP, and TP activate phosphatidylinositol metabolism, leading to the formation of inositol trisphosphate with mobilization of intracellular Ca^2+^ stores. Here we focus on the role of prostanoids in metabolic diseases.

#### 2.5.1. Prostaglandins

Prostaglandin I Receptor (IPR): IP receptors are found in leukocytes, T cells, platelets macrophages, pneumocytes, smooth muscle cells, and fibroblasts. PGI_2_ is the endogenous ligand for the IP receptor, mainly produced by vascular endothelial and smooth muscle cells, and inhibits platelet aggregation and thrombus formation [225,226]. PGI_2_ is primarily produced in mammalian vasculature with elevated levels in pulmonary arterial segments compared to the systemic circulation.

PGI2 activates adipogenesis by increasing the expression of C/EBPβ and C/EBPδ via the cAMP–PKA pathway and promotes adipocyte differentiation [227]. Deletion of PGIS and IP receptors significantly reduced body weight gain suppressed HFD-induced hypertrophy of adipocytes [228]. PGIS^−/−^ mice are protected from hepatic steatosis but not insulin resistance [229]. PGIS is expressed in the stromal vascular fraction and not in adipocytes, and HFD increases the expression of PGIS. PGI2 levels are decreased in obesity [230]. Beraprost, a PGI2 analog, suppressed the pathogenesis and development of diabetes and its complication, nephropathy, accompanied by improving glucose intolerance and insulin resistance in obese Zucker rats [231]. In obese rats, nitration of PGIS causes inhibition in the synthesis of PGI2 and is responsible for preventing functional hyperemia during exercise in skeletal muscle [230].

Polymorphisms in PGIS and the IP receptor are associated with essential hypertension [232]. Prostacyclin receptor variant (R212C) defective in adenylyl cyclase activation promotes increased platelet aggregation and atherothrombosis [233]. PGI_2_ limits pulmonary hypertension induced by hypoxia and systemic hypertension induced by Ang II [234]. PGI_2_ and its stable analogs were used successfully to treat pulmonary arterial hypertension [235]. Prostacyclin receptor knockout leads to intimal hyperplasia, atherosclerosis, and hypercoagulability as reperfusion injury and atherogenesis in mice [232,236,237,238]. PGI_2_ regulates both innate and adaptive immunity and affects the function of dendritic cells, macrophages, monocytes, endothelial cells, and eosinophils [239].

PGI_2_ role in cardiovascular health involves inhibiting platelet aggregation and vasodilatory effects via relaxation of smooth muscle. PGI_2_ analogs were successfully used for therapy in pulmonary arterial hypertension, peripheral occlusive disease, the vascular complication of diabetes mellitus, and treatment of reperfusion injury. In addition, in recent years, prostanoids were shown to have an important role in the resolution of inflammation.

Thromboxane Receptor (TP): COX1 activity increases thromboxane levels in activated platelets causing platelet adhesion and the risk of atherothrombosis [240]. In obesity, increased adipokines, such as leptin and adiponectin, are associated with platelet function. Knockout of either leptin or leptin-receptor protects from thrombosis in mice while adiponectin^−/−^ has increased thrombosis [241]. High adiponectin plasma concentrations are associated with a decreased risk of coronary artery diseases and increased bioavailability of NO [242]. Clinical studies correlating obesity to platelet aggregation are conflicting. Thromboxane A2, a marker of platelet activation, is higher in obese subjects than in lean subjects. However, insulin-sensitive morbidly obese subjects had lower levels of TBXB2 than the insulin-resistant obese subjects. Thus, leptin resistance combined with insulin resistance in a percentage of obese patients may influence variations in platelet function in obesity. TBXAS^−/−^ mice showed increased insulin sensitivity in mice fed a low-fat diet, not on HFD. On HFD, TBX^−/−^ mice had decreased inflammation and adipose tissue fibrosis [243].

An increase in thromboxane levels and a decrease in IP receptor levels may contribute to platelet hyperreactivity in humans with T2D [244]. An increase in adipokines resistin, leptin, PAI-1 and retinol-binding protein 4 in patients with metabolic syndrome and T2D induce insulin resistance in megakaryocytes by interfering with IRS-1 expression, therefore overcoming the inhibitory effects of insulin on platelets [245]. In poorly controlled diabetes, increased plasma levels of 8-iso-PGF2 due to increased lipid peroxidation also causes persistent platelet activation. PGI_2_ and TXA_2_ levels are increased in patients with atherosclerosis and ApoE^−/−^ mice. COX-2 is expressed by monocytes/macrophages in mouse atherosclerotic lesions and can increase TXA_2_ in atherosclerotic plaques and foam cells [246]. ApoE^−/−^ mice with TP deficiency showed a decrease in the extent of the atherosclerotic lesion with time [247]. TXA2 is produced by activated macrophages and is proinflammatory [248]. COX-1 deletion in bone marrow-derived cells decreases platelet thromboxane levels worsens early atherosclerosis in ApoE^−/−^ and LDLR^−/−^ mice [249].

COX inhibitors produce mixed results in mouse models of atherosclerosis. The use of NSAIDs selective for COX-2 inhibition increases the risk of a thrombotic event. Animal models suggest that targeting the TP may provide superior beneficial cardiovascular effects. Despite preclinical evidence, there is a limited indication for the superiority of TXA inhibitors, TP antagonists, or dual inhibitors of both targets compared with aspirin.

Prostaglandin E2(PGE2) PGE2 is a major prostanoid of AA metabolism and can bind to four receptors EP1–EP4. PGE2 exerts an anti-lipolysis effect in humans and mice and facilitates adipose tissue lipid accumulation [250,251]. However, the role of PGE2 production in the development of obesity and associated complications is not apparent. mPGES-1^−/−^ mice exhibited resistance to diet-induced obesity when compared to wild-type littermates [252]. They showed a lower body weight gain and reduced adiposity, and inflammation in adipose tissue. mPGES-1^−/−^ mice on HFD showed increased energy expenditures without any changes in activity and browning process. Altogether, these data suggest that mPGES-1 inhibition may prevent diet-induced obesity [228,253,254]. In addition, COX-2 and EP3 receptor inhibitors reversed obesity-induced tissues inflammation and obesity-linked complications [255].

EP3 mice develop a more robust obese phenotype on HFD [251,256]. Deletion of adipocyte phospholipase increases lipolysis, and the mice are resistant to diet-induced obesity [257,258]. EP3 is increased primary adipocytes isolated from the HFD-induced obese rats and human subjects, as well as 3T3-L1 and human adipocytes during the development of adipocyte hypertrophy and hypoxia [36]. Furthermore, in the genetically obese db/db mice, treatment with the EP3 antagonist significantly reversed the obesity-induced adipose tissue inflammation [259]. The 3T3-L1 adipocytes, treated with palmitate-and hypoxia, are hypertrophic and hypoxic and mimic the state in upregulated obesity. The blockade of COX-2 and EP3-mediated signaling suppressed MCP-1 and RANTES from these adipocytes [36]. EP4 signaling suppresses adipocyte differentiation and protects against the diabetogenic toxicity of streptozotocin in mice [260,261]. The treatment of the EP4 agonist in db/db mice decreased the levels of proinflammatory cytokines and chemokines and improved insulin sensitivity and glucose tolerance [261].

Further, EP4 activation increased the expression of adiponectin and peroxidase proliferator-activated receptors in white adipose tissue [261]. EP4^−/−^ mice fed HFD showed higher mRNA levels of EP2, EP3, and EP4 w in epididymal fat tissue [262]. The up-regulation of EP3 was accompanied by the downregulation of EP4 in the obese primary adipocytes isolated from the HFD-induced obese rats and human subjects [263]. PGE_2_ has been studied more extensively in vasculature beds, such as the renal circulation, and was shown to have four receptor subtypes, EP1–EP4. In the kidney vasculature, EP2 and EP4 stimulation result in vasodilation, whereas EP1 and EP3 stimulation result in vasoconstriction [253]. Cardiomyocyte-specific deletion of the EP4 exacerbates the decline in cardiac function after myocardial infarction [232,264]. These observations raise the possibility that, despite results in healthy volunteers, inhibition of mPGES-1 in male patients with hyperlipidemia may predispose them to an exaggerated BP response to an HSD [265]. While human studies have shown that during acute and prolonged exercise, PGE_2_ levels increase in the blood, the role of this prostaglandin in skeletal muscle blood flow control is unclear [266,267,268,269]. PGE_2_ may also play a role in other vasculature beds, such as the cerebral circulation [270]. Further work to elucidate the role of PGE_2_ and the EP receptors in skeletal muscle blood flow control is warranted.

PGE2 is a key mediator of the inflammatory process in the cardiovascular system proinflammatory cytokines, and NO upregulates PGE2 synthesis in smooth muscle cells and macrophages [254,271]. PGE2 is increased in patients with acute coronary syndromes and associated with adverse clinical outcomes (myocardial infarction or death) at follow-up elevated in patients. Experimental studies in mice have demonstrated that cox’s potentially deleterious effects in humans are due to endothelial and vascular SMC. COX-2 is responsible for most of the PGI2, and therefore, inhibition of COX-2 functions results in mice in a hypertensive and prothrombotic phenotype. The prothrombotic phenotype of endothelial/vascular SMC COX-2 deletion was also observed in IP receptor knockout mice [240].

COX-2, mPGES-1, EP-3, and EP-4 mRNA expression and PGE2 levels are increased in PBMC of patients with coronary artery disease than healthy volunteers, suggesting that circulating monocytes are activated [272]. Thus, COX-2-mediated PGE_2_ overproduction by stimulated monocytes might provide a new marker of subclinical atherosclerosis in asymptomatic subjects with cardiovascular risk factors [273].

Patients with carotid atherosclerosis overexpress COX-2, mPGES-1, and EPs simultaneously in the PBMC and the vulnerable region of plaques. The studies in cultured monocytic cells suggest that NF-κB inhibitors and/or EPs antagonists could represent a novel therapeutic approach to treating plaque instability and rupture. PGE2 suppresses the production of proinflammatory cytokines and chemokines via EP4 in LPS–treated human and murine macrophages [274]. EP4 activation suppresses chronic inflammation in vivo by mitigating macrophage activation during ischemia-reperfusion injury, atherosclerosis, allograft rejection after cardiac transplantation, and abdominal aortic aneurysm [275].

Prostaglandin F_2α_ (PGF2): PGF_2_*_α_* is synthesized by PGF synthase (PGFS) enzymes including Aldo ketoreductase(Akr) and prostamide/PGFS [276]. F prostanoids are an emerging class of bioactive lipids; also, products of the AA metabolism formed not via the classical COX pathways but a free radical-mediated mechanism [277]. During the past decade, these chemically stable prostaglandin isomers, generally called F2-isoprostanes (F2-iPs), have emerged as reliable and sensitive measures of in vivo lipid peroxidation and oxidative stress [278]. Akr1C3 acts as a PGFS in adipocytes and is associated with suppressing adipogenesis through inhibition of PPAR*γ* function [279]. Thus, PGF_2_*_α_* suppresses an early phase of adipogenesis. Fluprostenol, an FP receptor agonist, reduces the expression of PPAR*γ* and its target genes suppressing adipogenesis, which can be reversed by treatment with AL8810, an FP receptor antagonist [279]. Akr1B7 gene-knock-out mice display excessive adiposity resulting from adipocyte hyperplasia/hypertrophy and exhibit high sensitivity to diet-induced obesity. Treatment of 3T3-L1 cells or AKR1B7 gene-knock-out mice with FP receptor agonists decreases adipocyte size and inhibits the expression of lipogenic genes.

The FP is expressed in pre-glomerular arterioles, renal collecting ducts, and the hypothalamus. PGF2 dose-dependently elevates blood pressure in WT mice via activation of the F prostanoids (FP) receptor [280]. Deleting the FP reduces blood pressure, coincident with a reduction in plasma renin concentration, angiotensin, and aldosterone, despite a compensatory up-regulation of AT1 receptors and an augmented hypertensive response to infused angiotensin II [279]. Atherogenesis is attenuated by deletion of the FP, although the receptor is not expressed in the aorta or atherosclerotic lesions in LDLR^−/−^ mice [279]. FP/LDLR double KO mice have decreased vascular TNF_α_, inducible nitric oxide enzyme, and TGF_β_ and reduced macrophages in lesions. Its deletion does not alter macrophage cytokine generation [281]. Thus, blockade of the FP offers an approach to the treatment of hypertension and systemic vascular disease.

Vascular oxidative stress increases the generation of free radicals and lipid oxidation products, a key element in atherogenesis [282]. In hypercholesterolemic patients, elevated concentrations of F2-IP(F2-isoprostanes) correlate with cholesterol levels and decrease with statin therapy. F2-IPs are elevated in people with diabetes predisposed to accelerated atherogenesis [283]. The increase in isoprostanoids also occurs in different mouse models of genetic hypercholesterolemia and atherogenesis, and antioxidants reduce both their levels and the development of the disease [284].

#### 2.5.2. Leukotriene

LTB4/BLT1/BLT2. BLT1 is a Leukotriene receptor (BLT)1 and is expressed in leukocytes, including granulocytes, T Cells, dendritic macrophages, and vascular smooth muscle cells [285]. Leukotriene B_4_ (LTB_4_) is a potent proinflammatory mediator derived from arachidonic acid via the 5-lipoxygenase pathway and is produced by PMN. LTB4 binds to BLT1 with high affinity and to BLT2 with low affinity to induce inflammation. BLT2 was originally reported as a low-affinity LTB4 receptor and is identified as a receptor for oxidized fatty acids [286].

Both 5-lipoxygenase and LTB4 levels are increased in the liver and adipose tissue in murine models of experimental obesity and HFD fed rodents [287]. In addition, 5-LO^−/−^ mice and mice treated with LTB4 antagonists are protected from HFD-induced insulin resistance and show decreased macrophages and T cells infiltration in adipose tissue [288]. Similarly, inhibition of the 5-lipoxygenase pathway in obese mice reduced proinflammatory cytokines and circulating free fatty acid concentrations, reversed insulin resistance and hepatic steatosis [289].

BLT-1^−/−^ mice have decreased inflammation and macrophage accumulation in adipose tissue and are protected from the development of insulin resistance in diet-induced obesity (DIO). BLT-1 deletion in ob/ob mice decreased hepatic triglyceride accumulation and inflammation and had beneficial effects on hepatic steatosis and nonalcoholic fatty liver disease [290]. In obese mice, increased uptake of omega-3-polyunsaturated fatty acids enhanced insulin sensitivity and anti-inflammatory mediators such as resolvins and protectins in adipose tissue and decreased LTB4 [143,291]. LTB_4_^−/−^ mice were protected mice from diet-induced insulin resistance. Inhibition of LTB4 synthesis or treatment with BLT1 antagonists in T1D and T2D diabetes reduced inflammation in adipose tissue in obese mice [292].

Subjects with features of the MetS have lower stimulated LTB4, which is not due to increased metabolism of LTB4. Weight reduction restored the production of neutrophil LTB4, suggesting that in addition to modifying cardiovascular risk, weight loss may also help manage inflammatory responses in overweight subjects [293]. LTB4 inhibition reduced lipolysis in adipose tissue and plasma levels of FFA in diet-induced obese mice [294]. The LTB4/BLT1 is implicated in recruiting B2 cells to the adipose tissue of obese mice, leading to T cells activation and insulin resistance [295]. B-cell null mice do not develop HFD induced insulin resistance. However, the adoptive transfer of adipose tissue B2 cells from wild-type HFD donor mice into HFD B^−/−^ mice restored the effect of HFD to induce insulin resistance [296].

In atherosclerosis, LTB_4_ increased MCP-1 secretion and adhesion of monocytes to endothelial cells. In LDLr^–/–^ and ApoE^–/–^ mice. The BLT antagonist CP-105,696 and knockout of BLT1 in ApoE^−/−^ mice protected from atherogenesis [297]. BLT^−/−^ with decreased expression of CD36 (a fatty acid translocase, B-type scavenger receptor) and CCL2 chemokine, and by the reduced recruitment of smooth muscle cells to the atherosclerotic lesions. Inhibition of BLT1 receptor with CP-105,696 reduced arterial pressure in the SHR compared to the normotensive control, and inhibitors of 5-LO prevent the development of PAH in animal models. LT synthesis—5-LO, FLAP, and LTA4 hydrolase—are expressed in the lung vessels from patients with severe PAH.

#### 2.5.3. Hydroxy Eicosatetraenoic Acids

Cytochrome P450-mediated AA metabolites have a significant role in normal physiological and pathophysiological conditions; hence they could be promising therapeutic targets in different disease states. P450 monooxygenases mediate the (ω-n)-hydroxylation reactions, which involve introducing a hydroxyl group to the carbon skeleton of AA, forming subterminal hydroxy eicosatetraenoic acids (HETEs). The 20-HETE is converted to 20-hydroxy-prostaglandin G2 and H2 by cyclooxygenase and promotes vasoconstriction [298].

GPR75/20-HETE: Arachidonic acid can be oxidized by several cytochrome P450 mixed-function oxidases to produce various HETEs [299]. The 20-Hydroxyeicosatetraenoic acid (20-HETE) is the omega-hydroxylated metabolite of arachidonic acid produced by the cytochrome P450 (CYP) 4A12 and 4F enzymes [300]. GPR75 binds 20-HETE and promotes vascular smooth muscle contraction, endothelial dysfunction, inflammation, and cell proliferation [301]. The 20-HETE is increased in individuals with obesity (BMI > 30) and metabolic syndrome [302] and animal models of obesity and by HFD. Polymorphism in the human 20-HETE synthase CYP4F2 is associated with metabolic syndrome phenotypes [303,304].

CYP4A proteins are upregulated in livers of mice with genetically induced and diet-induced diabetes [305]. Inhibition of CYP4A in mice reduces hepatic ER stress, apoptosis, insulin resistance, and steatosis. CYP4A14 knockout male mice, a model for 20-HETE, had increased weight gain and metabolic syndrome hyperglycemia and diabetes, including diabetic retinopathy and nephropathy [306]. The 20-HETE antagonist attenuated weight gain and prevented the development of insulin resistance in these mice [302]. Similar results were obtained in male and female transgenic mice that overexpress the 20-HETE synthase CYP4A12 on HFD. The 20-HETE antagonist, 20-SOLA, attenuated weight gain and prevented the development of hyperglycemia and impaired glucose metabolism. Inactivation of IR and IRS-1 was identified as the mechanism for insulin resistance [307]. Additional studies in 3T3-1 differentiated adipocytes confirmed that 20-HETE impairs insulin signaling and that its effect may require activation of its receptor GPR75 [307]. Therefore strategies to reduce levels or activity of CYP4A proteins in the liver might be developed to treat T2D [305].

Clinical studies have shown elevated plasma, and urinary 20-HETE in hypertension, obesity and metabolic syndrome, myocardial infarction, stroke, and chronic kidney diseases [308]. Mutations in CYP4A11 and CYP4F2 are associated with the development of hypertension. Studies in CYP4A14 KO and inducible CYP4A12 transgenic and DHT-treated mouse models indicate increased vascular 20-HETE production, and these mice are hypertensive [309,310]. In mice, 20-HETE activation of GPR75 contributes to the development of hypertension knockdown of the expression of GPR75 mimics the effects of 20-HETE inhibitors to prevent the development of hypertension and vascular hypertrophy in a CYP4A12 transgenic mouse model [311]. These findings imply that GPR75 may be a viable target for the treatment of hypertension.

GPR31/12-HETE 12/15-LOX, predominantly expressed in macrophages and pancreatic islets in mice, catalyzes the conversion of arachidonic acid to eicosanoids 12-hydroxyeicosatetraenoic (12-HETE) and 15-hydroxyeicosatetraenoic acid (15-HETE) [312]. The 12-HETE mediates its effects through several receptors, including the GPR31 and low-affinity leukotriene B4 (BLT2) receptor. Protons and lactic acid also activate GPR31 [313]. The 12-HETE generation increases oxidative stress and modulates inflammation via interaction with GPR31 and its low-affinity receptor BLT2.

The 12/15-LOX isoforms are expressed in adipose tissues from patients with obesity, particularly in the stromal vascular fraction along with inflammatory cells such as macrophages. In addition, 12-HETE promotes proinflammatory cytokines and chemokines, such as TNF-α, MCP-1, and IL-6 in adipocytes.

The 12-LO expression in pancreatic islets increases during metabolic stresses, such as hyperglycemia, cytokine-mediated damage, and partial pancreatectomy. The 12-HETE acts via GPR31 in promoting β-cell dysfunction in the setting of insulin resistance and inflammation in both macrophages and pancreatic islets [314,315]. It is also essential for pancreatic organogenesis [316]. Recent studies show that 12-LO^−/−^ mice fed an HFD exhibit reduced macrophage infiltration into adipose tissue, reduced insulin resistance, enhanced β cell function, and improved glucose tolerance compared to controls [317,318]. Pancreatic deletion of 12-LO protects obese HFD fed mice from glucose intolerance and improves insulin secretion in cytokine-treated islets in a 12-HETE-dependent manner [319]. Deletion of 12-LO in adipocytes driven by the aP2-Cre transgene protects mice from HFD-induced glucose intolerance. Taken together, 12-HETE appears to have a prominent role in DIO inflammation, insulin resistance, and glucose intolerance and is suitable for treating obesity and diabetes.

The 12-LOXs promote atherosclerosis by LDL oxidation, and induction of a proinflammatory state enhances macrophage metabolic activity. The 12(S)-HETE proinflammatory effect induces monocyte binding to human aortic endothelial cells, promotes endothelial wall disruption, and directly oxidizes LDL, which contributes to foam cell formation [320]. Mice with Alox15^−/−^ on the ApoE^−/−^ background developed significantly reduced atherosclerotic lesions even at one year of age. Deleting Alox15^−/−^ in the LDLr^−/−^ or ApoE^−/−^ mice leads to a significant reduction in plaque formation after HFD.

Binding of 12(S)-HETE to GPR31 on platelets leads to increased thrombosis in the mouse carotid injury model. In endothelial cells, 12(S)-HETE binding causes the release of ADAMTS-18, which binds to platelets and causes the release of HETE and platelet fragmentation [321]. The GPR31 pepducin inhibitor effectively inhibited occlusive arterial thrombosis without detectable effects on hemostasis in animal models. This suggests that 12(S)-HETE-GPR31 could be a new antithrombotic and anti-stroke target [322]. A comprehensive review on lipoxygenases was published recently, and the readers are referred to these publications [314,322].

## 3. TCA Cycle Metaboltes

TCA cycle metabolites are byproducts of cellular metabolism necessary for the biosynthesis of macromolecules such as nucleotides, lipids, and proteins. Alterations in the TCA cycle have correlated with numerous pathologies, including cardiovascular diseases and metabolic syndrome, where mitochondrial function and oxidative stress play a key role [323,324,325,326]. In addition, emerging evidence indicates that TCA cycle metabolites have systemic effects and function as messengers between different metabolic organs [327,328,329].

GPR91/SUCNR1 GPR91 is expressed in white adipose tissue, liver, heart, retinal neurons, intestine, spleen, and immune system cells, including dendritic cells and couples Gi/o and Gq-depending on the tissue [330]. Succinate is released from mitochondria during cell damage, hypoxia, free-radical processes, mitochondrial dysfunction, and uncoupling of oxidative phosphorylation [320,331]. Consequently, elevated amounts of circulating succinate occur in physiological conditions, such as endurance exercise and specific pathologies, including hypertension, ischemic heart disease, T2D, and obesity [332,333,334,335]. GPR91 expression was identified in an adipose cluster, with a high level in white adipose tissue (WAT) and abundant amount in purified adipocytes, which enables the extracellular succinate to downregulate lipolysis.

High succinate was detected in spontaneously hypertensive rats (SHR), ob/ob mice, db/db mice, and fa/fa rats compared to their non-diseased controls [336,337]. GPR91 expression is high in white adipose tissue (WAT) and purified adipocytes, enabling the extracellular succinate to downregulate lipolysis when glucose and free fatty acid molecules are present in excess [338]. GPR91^−/−^ mice on HFD showed a decrease in macrophage infiltration into adipose tissue and improved glucose tolerance with no difference in body weight compared to WT mice [335,339]. Activation of GPR91 in liver tissue has unfavorable effects on NAFLD. However, increasing this signaling axis in white adipose tissue improves liver lipotoxicity in an obesogenic setting. In patients with obesity and T2D, increased levels of succinate correlate with increased BMI, insulin, glucose, insulin resistance, and triglycerides. In contrast, BAT, which has excess mitochondria, is the primary succinate-metabolizing tissue [338]. GPR91deletion in myeloid cells protected mice from obesity on HFD, but these mice showed impaired glucose tolerance and insulin sensitivity [335,340,341]. GPR91^−/−^ myeloid cells had decreased anti-inflammatory response to type 2 cytokines, including those associated with diet-induced obesity [340].

In the heart, GPR91 mRNA and protein are localized in the sarcolemmal membrane and the T-tubules. Succinate increases cardiac output in ischemia and hypoxia, and the receptor is suggested to have a regulatory role in the heart [342,343]. High succinate was detected in spontaneously hypertensive rats, ob/ob mice, db/db mice, and fa/fa rats compared to controls [333]. Intravenous administration of succinate into mice or humans causes elevation of blood pressure which was eliminated by treatment with captopril [333]. In vitro and in vivo succinate causes cardiac hypertrophy and was eliminated in GPR91-KO mice. Prolonged incubation of cardiomyocytes with high succinate concentrations induces apoptosis [330]. GPR91 was upregulated in the hearts of pulmonary banding rats and human RV hypertrophy [344] In platelets, succinate induces platelet aggregation via an increase in the activity of IIb/IIIa receptors [327].

GPR91 is expressed on DCs, mast cells, bone marrow-derived macrophages, adipose tissue macrophages. The functional effects of GPR91 activation in innate immune cells are both cell and context-dependent. In immature DCs, succinate stimulates cell migration in a concentration-dependent manner and thus mediates chemotaxis [336,345,346]. SUCNR1 expression is induced during the development of immature DCs from monocytes. SUCNR1 and toll-like receptors act in synergy to potentiate the production of the inflammatory cytokines tumor necrosis factor α (TNFα) and interleukin [347]. SUCNR1 activation increases the intracellular release of arachidonic acid that, through the actions of cyclooxygenase (COX)-2, leads to the production and release of prostaglandin that subsequently transactivates EP2 and EP4 receptors on the granular cells [348]. Extracellular succinate increases the expression and release of VEGF under hypoxic conditions [330].

GPR91 has value as a potential therapeutic target based on the regulatory roles succinate plays in lipid metabolism, blood cell and vessel formation, blood pressure and the cardiovascular system, and immune responses [349,350]. Thus, there is significant interest in finding agonists and antagonists of GPR91 as potential substances for the pharmacotherapy of hypoxic disorders, renal hypertension, diabetic lesions, metabolic syndrome, autoimmune diseases [351]. A better understanding of the mechanisms controlling and regulating metabolic functions in health and pathology is required to develop new pharmacological strategies to prevent and treat these disorders.

GPR99/α-ketoglutarate receptor (AKG) The GPR99 receptor is also known as GPR80, OXGR1, P2Y15, and AKG and binds the TCA cycle metabolite alpha-ketoglutarate. GPR99 a Gα_q_-coupled GPCR binds the TCA cycle metabolite, α-ketoglutarate (AKG), but the physiological function is not clear [352]. GPR99 is expressed in the brain, lung, kidney, heart, and skeletal muscle [353]. Dietary α-KG would inhibit weight gain in male and female mice fed with a regular chow or HFD [354]. Accumulation of α-ketoglutaric acid prevents diet-induced obesity by adrenal activation of adipose tissue thermogenesis and lipolysis [355].

Increased expression of GLUT1 in diabetic rats increases glycolysis and accumulation of TCA metabolites succinate and αKG [356]. STZ-induced type I diabetic rats show increased urinary levels of AKG, citrate, and succinate. In the kidney, high glucose conditions promote increased intratubular AKG and OXGR1-dependent AngII formation and Na^+^ reabsorption [357].

α-ketoglutaric acid levels in plasma correlate to the risk of cardiovascular diseases and are connected to an early-onset inherited risk of stroke. When exogenously expressed, it activates the proliferation of fibroblasts. *GPR99* KO mice show a significant increase in cardiac hypertrophy decrease in cardiac shortening and ejection fraction following transverse aortic constriction [358,359]. Neonatal rat cardiomyocytes overexpressing OXGR1 show reduced phenylephrine-induced cardiomyocyte hypertrophy [360].

α-ketoglutarate modulates inflammation by promoting an M2 macrophage phenotype [361]. In addition, in respiratory cells, it binds leukotrienes and promotes inflammation, and vascular leak [362], Conversion of AKG to glutamine serves as a fuel for immune cells.

In addition, binding GPR99 to multiple ligands such as leukotrienes and AKG may complicate its utility as a therapeutic target. AKG is also shown to have antioxidant effects and has recently been shown to reverse aging. However, future studies must identify all-natural receptor ligands and determine their tissue-specific effects before being used therapeutically.

## 4. Amino Acid Metabolites

Amino acids are the backbones of cellular proteins and contribute to synthesizing other metabolites such as purine/pyrimidines and neurotransmitters [363]. In addition, amino acid-derived metabolites activate four GPCRs: GPR142, Calcium-sensing receptor (CaSR), Trace amine-associated receptor 1 (TAAR1), and GPR35. Although other amino acid metabolites also influence metabolism, we focus on the amino acid metabolites that bind GPCRs and influence metabolic disease [364,365].

GPR142/Tryptophan: GPR142 is a GPCR expressed in the pancreas and the immune system and shares 33% amino acid identity with GPR139 [366]. Recently, ligands for GPR139 were reported as being the essential amino acids L-tryptophan and L-phenylalanine. GPR142 binding of L-Trp triggers the activation of both Gq and Gi-coupled signaling and the activation of ERK [367].

Dietary polypeptides and amino acids stimulate insulin and incretin hormone secretion and regulate postprandial glycemia in animals and humans. Aromatic amino acids such as tyrosine (Tyr), phenylalanine (Phe), and tryptophan (Trp) are elevated in the blood plasma of insulin-resistant and diabetic patients [11]. GPR142 levels were increased during fasting and decreased in DIO. Tryptophan binding to GPR142 increased GSIS in lean mice, DIO mice, and obese mice. However, KO studies showed contributes to the augmented GSIS by tryptophan in obese animals [368]. GPR142 agonist did not affect body weight in DIO mice, but increased energy expenditure and carbohydrate utilization lowered basal glucose and improved insulin sensitivity [366]. In a small study with T2D, tryptophan delayed the rise in blood glucose after a carbohydrate meal by slowing gastric emptying response [369].

In diabetic long-term feeding with tryptophan-enriched chow delayed the onset and progression of diabetes, and it is assumed that tryptophan protected the pancreatic β-cells from exhaustion, are anti-inflammatory, and reduced the absorption of glucose from the small intestine [370,371]. In addition, GPR142 agonists also increased β-cell proliferation, which they interpreted to be an indirect effect mediated through local production of GLP-1 in the islets. Thus, GPR142 agonists could potentially modify metabolism through a balanced action of gut hormones as both insulin and glucagon and is a novel therapeutic approach for treating diabetes with minimal risk for hypoglycemia which has led to the design of synthetic GPR142 agonists, which have recently reached phase 1 in clinical trials for Type 2 diabetes treatment [372,373].

GPR35/Kynurenic acid receptor: GPR35 is a G_i_ and Gα_13_ coupled orphan GPCR that binds kynurenic acid (KYNA), a catabolite of tryptophan. KYNA is generated by the irreversible transamination reaction between l-KYN and 2-oxoacid by kynurenine aminotransferases. It is expressed in several tissues, including the digestive tract, skeletal muscle, lung, liver and heart, and immune cells [374,375,376,377,378]. GPR35 is present in pancreatic islets and skeletal muscle, with relatively higher levels in the adult lung, small intestine, colon, and stomach [379,380].

GPR35 stimulates lipid metabolism, thermogenic, and anti-inflammatory gene expression in adipose tissue [381,382]. Kynurenic acid suppresses weight gain in animals fed an HFD and improves glucose tolerance. Treatment of mice with Kynurenic acid in the DIO model reduced body weight, inguinal WAT mass, and improved glucose tolerance and plasma triglyceride levels. In addition, kynurenic acid and GPR35 enhance Pgc-1α expression and cellular respiration and increase the levels of Rgs14 in adipocytes, which leads to enhanced beta-adrenergic receptor signaling. GPR35^−/−^ mice exhibit progressive weight gain and glucose intolerance and sensitize to the effects of high-fat diets. Finally, exercise-induced adipose tissue browning is compromised in GPR35 knockout animals [382]. GPR35 agonists could thus be effective as an anti-obesity target [383].

GWAS has identified GPR35/CXCR8 SNP that was associated with diabetes [384]. Kynurenic acid levels are increased in the peripheral blood of patients with T2D Agonists for GPR35 reduced blood glucose levels in oral glucose tolerance tests, stimulated glucose uptake in differentiated 3T3-L1 adipocytes, and reduced free fatty acid plasma levels in both fasted wild type and diabetic (db/db) mice. A GPR35 expression was observed in the pancreas of db/db mice but not obese (ob/ob) diabetic mice using quantitative polymerase chain reaction. The adipose, liver, spleen, and colon expression levels remained similar between these two transgenic lines. Thus, GPR35 may play a role in glucose uptake, storage, and transport. However, more studies are required to probe the role of GPR35 in the mediation of glucose homeostasis and diabetes [381].

GWAS studies implicate GPR35 in the pathology of atherosclerotic plaque formation and coronary artery disease risk in a patient cohort [381]. In a deoxycorticosterone acetate-salt induced hypertensive model, male GPR35 knockout mice were protected from hypertension with improved endothelium-dependent vasodilation and decreased superoxide in isolated aortas [385]. A previous report showed an increased BP profile in GPR35 knockout mice under anesthesia compared with the wild-type controls [20]. Yet, in another study, mice with GPR35 deletion showed resistance to BP elevation in Ang II-induced hypertension [376,386]. Inhibition of GPR35 preserves mitochondrial function after myocardial infarction by targeting Calpain 1/2 [387]. GPR35 gene and protein expressions were induced in mouse models of cardiac failure, the acute phase of myocardial infarction, and during the compensatory and decompensatory phase of pressure-load-induced cardiac hypertrophy [388]. Microarray analyses on heart failure patients showed that GPR35 was increased in heart failure [389]. While several endogenous ligands have been identified, GPR35/CXCR8 has recently been observed to bind the chemokine CXCL17. These results suggest that multiple tyrosine metabolites are alternative endogenous ligands of GPR35 and may represent a druggable target for treating certain diseases associated with the abnormality of tyrosine metabolism.

Emerging evidence indicates that kynurenine regulates immune system functions and inflammation [390]. GPR35 is expressed by human immune cells, including monocytes (CD14^+^), T-cells (CD3^+^), neutrophils, and various dendritic cells, and natural killer T cells (CD56^+^) and has a broadly similar expression pattern in mice [381]. GPR35/CXCR8 promotes the adhesion of leukocytes to vascular endothelium [384]. Several in vitro studies using various primary or immortalized leukocyte cell types have revealed that KYNA can attenuate inflammation elucidated by different stimuli. KYNA limits inflammation by decreasing oxidative stress.

Since KYNA has opposing tissue-specific effects, while beneficial to adipose tissue, it is elevated in IBD and neuropsychiatric disorders [391]. However, KYNA affects diverse immune-related signaling pathways and requires further in-depth analysis to avoid unexpected adverse consequences. GPR35 has a significant role in inflammatory pain, asthma, diabetes, hypertension, cardiovascular disease, and irritable bowel disease. Another consideration is that KYNA also binds to the aryl hydrocarbon receptor, which may influence the outcome. Before being translated to clinics, further studies to characterize the species selectivity of ligands and their role in different diseases will be required [381].


Calcium-Sensing Receptor (CaSR)/phenylalanine


The calcium-sensing receptor (CaSR) is a GPCR involved in calcium homeostasis and couples to G_q/11_, G_i/o_, and G_12/13_ and Gα_s_ proteins [367]. It is expressed in kidney and parathyroid glands and to a lesser extent in lungs, skin, intestine, brain, and vasculature [392]. Its physiological endogenous ligands include the polyamines putrescine, spermidine, and spermine, the aromatic amino acids, including L-phenylalanine and L-tryptophan, as well as poly-L-arginine and β-amyloid peptide. The binding of phenylalanine to CaSR sensitizes the binding of Ca^2+^ to the receptor. Studies of CaSR mutations, which cause disorders of calcium homeostasis, showed. G-protein independent signaling [393]. Functional assessment of these mutations demonstrated the importance of the homodimer interface and transmembrane domain in biased signaling. CaSR regulates vascular tone, metabolic processes in vascular cells, lung and neuronal development, or cardiac function [392].

Oral administration of L-Phe acutely reduced food intake in rats and mice and chronically reduced food intake and body weight in diet-induced obese mice [394]. The anorectic effects of L-Phe are mediated via the CaSR and suggest that L-Phe and the CaSR system in the GI tract may have therapeutic utility in treating obesity and diabetes [394]. CaSR can mediate the inhibition of lipolysis in adipocytes [395]. Phenylalanine is higher in obese subjects than in normal controls [396]. Phenylalanine levels decrease after performing bariatric surgery [397]. A high prevalence of overweight and obesity was observed in females with phenylketonuria [398]. Patients with elevated phenylalanine levels had substantially more incompletely metabolized waste of fatty acids in the circulation owing to impaired mitochondrial β-oxidation, indicating dysfunctional energy production machinery [399]. 

CaSR is required for hormone secretion in the specific response to L-Phe by the native L-cell and is expressed by pancreatic β-cells and can promote glucose-induced insulin secretion [400].

CaSR polymorphisms are associated with coronary artery diseases (CADs) such as myocardial infarction and atherosclerosis [401]. Elevated phenylalanine levels predicted mortality in heart failure patients, independent of traditional prognostic factors and cytokines associated with inflammation and immunity [402]. Higher phenylalanine levels correlated with higher C-reactive protein levels and higher pro-inflammatory, innate, and adaptive T lymphocytes immune cytokines such as IL-8 and IL-10. Inflammation increases phenylalanine levels in patients with HF [265]. The leucine/phenylalanine ratio could be a valuable predictor of future cardiac events in patients with HF, reflecting an imbalance in amino acid metabolism [403]. Phenylalanine levels are associated with pulmonary hypertension in metabolic profiling clinical studies and suggested as a therapeutic option [404,405].

CaSR is expressed on immune cells, such as monocytes, macrophages, proerythroblasts, erythroblasts, and megakaryocytes [393]. CaSR is also expressed by T lymphocytes, although not by B lymphocytes [406]. Elevated phenylalanine levels in inflammation in HIV infection, burn patients, sepsis, and higher Phe/Tyr correlate with the clinical course and predict non-survival. In addition, CaSR activates NLRP3 inflammasome, amplifying the inflammation response mediated by increased intracellular inositol phosphate/Ca^2+^ pathway in monocytes and macrophages [407]. Newer studies of CaSR show evidence of tissue-specific regulation and endosomal signaling [408].

GPR139: GPR139 is still classified as an orphan receptor, as its endogenous ligand and function are unknown. As L-Trp and L-Phe are also putative endogenous ligands of GPR139, it is a nutrient-sensing receptor [409]. GPR139 homolog GPR142 shares the activation by L-Trp and L-Phe but is mainly expressed in the pancreas and gut, where it regulates insulin and incretin secretion, respectively, making GPR142 a potential target in type II diabetes. GPR139 is expressed in the hypothalamus, pituitary, and habenula in humans and rats and may regulate food consumption and/or energy expenditure [409]. Thus, based on the expression pattern and nature of the putative endogenous agonists, GPR139 could be involved in metabolism-related disorders such as T2 diabetes [410].

Furthermore, in the patent from Regeneron Pharmaceuticals, it is stated that the GPR139 knockout mice had increased lean body mass and decreased body fat compared to wild-type mice suggesting that it may have a role in energy homeostasis [409]. Additionally, appetite-regulating hormones ACTH, α-MSH, and β-MSH can activate GPR139 at relatively high concentrations [411,412]. Furthermore, GPR139 modulates the signaling of the canonical melanocortin receptors further supports the role of GPR139 in energy homeostasis [413]. Thus, GPR139 could be a potential target for treating metabolism-related disorders such as obesity and type II diabetes. The amino acids L-Trp and L-Phe and derivatives of the peptide hormones ACTH and α-MSH were suggested as potential endogenous GPR139 receptor agonists but remain fully validated.


Trace amine activated receptors (TAAR1)


TAAR1 is an intracellular amine-activated G_s_-coupled and G_q_-coupled GPCR primarily expressed in neuronal cells and peripheral organs such as the GI tract immune cells [414]. There are nine *TAAR* genes in humans, including three pseudogenes; nine genes in chimpanzees, including six pseudogenes; 19 and 16 in rats and mice with two and one being pseudogenes, respectively [415]. Endogenous agonists for TAAR1 include common biogenic amines and include β-phenylethylamine (β-PEA), tyramine, octopamine, tryptamine, and thyronamine [416]. Tyramine is a trace amine naturally found in multiple dietary sources: aged cheeses, aged meat, alcoholic beverages, chocolate, some fruits, and vegetables [417]. Tyramine is produced by decarboxylation of dietary amino acids and metabolized by the intestinal microbiota [418].

TAAR1 in pancreatic islets increases insulin secretion and glucose tolerance in mouse models [419]. TAAR1 activation by agonist increased glucose-dependent insulin secretion in INS1E cells and human islets and elevated plasma PYY and GLP-1 levels in mice [419]. In diabetic db/db mice, the TAAR1 agonist normalized glucose excursion during an oral glucose tolerance test [418]. Tyramine lowers the hyperglycemic response to a glucose load by stimulating glucose uptake in all insulin-sensitive tissues, including adipocytes and skeletal and cardiac muscle [420]. In the gut, it promotes motility, satiety, and eating behaviors. TAAR1 agonist decreased food intake and body weight in a diet-induced model of obesity with improved insulin sensitivity and plasma triglyceride levels [421]. Tyramine inhibited glycerol release by rat adipocytes enhanced fat deposition in epididymal white adipose tissue [422]. Tyramine concentrations were significantly reduced in patients with MetS and inversely correlated with multiple biomarkers of inflammation and cardiometabolic risk factors such as body mass index and blood pressure [423]. Therefore TAAR1 with an incretin-like mechanism could be a new target for treating T2D and obesity [419,424].

Tyramine increases blood pressure by release of noradrenaline, vasoconstriction, and increasing cardiac output [425]. Tyramine infusion causes a rise in systolic blood pressure in hypertensive individuals than in normotensives [426]. However, vasoconstrictor or vasorelaxant effects of trace amines may be tissue or vascular bed specific and needs further studies [427].

TAAR1 expression in immune cells includes human peripheral mononuclear cells, B lymphocytes, monocytes, polymorphonuclear leukocytes, NK cells, and T lymphocytes [428,429,430]. TAAR1 is upregulated in BMDM by different agonists, and tyramine increases inflammatory cytokine gene expression in non-polarized and LPS-polarized BMDM [431]. Tyramine, released from activated platelets, is speculated to be a chemotactic TAAR1 ligand for neutrophils [432]. Tyramine is cytotoxic activity to B cells expressing TAAR1 [433]. TAAR1/TAAR2 enhances Th2 responses by increasing IL-4 production TAAR1- and TAAR2-mediated IgE secretion is induced by biogenic amines in B cells [414].

Pre-clinical animal models have identified TAAR1 as a novel target for metabolic disorders and in regulating immune function. Thus, TAAR1 agonism may be a novel therapeutic strategy for treating T2D and also shows potential for the pharmacotherapy of obesity from both drug- and diet-induced causes. While at least some of the effects described above almost certainly arise from local effects, a role for TAAR1 in the CNS control of energy metabolism and nutrient intake should also be considered. Further, the recent demonstration of the ability of TAAR1 agonists to prevent binge eating allows such compounds to address both the centrally mediated over-consumption and subsequent insulin resistance and hormone imbalance aspects of obesity and associated metabolic disorders [434].

## 5. Nucleotide-Nucleoside Metabolites

ATP is produced from simple and complex sugars as well as from lipids via redox reactions. Carbohydrates are broken down into simple sugars, while the lipids are into fatty acids and glycerol. These substrates in mammalian cells are used to generate ATP by either mitochondrial oxidative phosphorylation or cytoplasmic glycolysis. Extracellular nucleotides, such as ATP, ADP, UTP, UDP are released into the extracellular milieu and blood from endothelial cells, erythrocytes, aggregated platelets, and activated leukocytes in response to hypoxia, oxidative stress, increased blood flow, mechanical and proinflammatory stimuli, cell damage, or death [435,436,437,438,439]. Extracellular nucleotides are degraded by membrane ectonucleotidases (ATPase and AMPase), CD73, and CD39 ATP metabolizing enzymes [437,440,441]. Extracellular nucleotides bind purinergic receptors, consisting of P1 receptors stimulated by adenosine and P2 receptors that bind extracellular nucleotides (ATP, ADP, UTP, and UDP) [442]. P1 and P2 receptors are expressed in the cardiovascular system, lungs, skeletal muscle, brain, kidneys, immune system, pancreas, and adipose tissue. Changes in nucleotide metabolism in diabetes, obesity, and insulin resistance were observed and need further studies to understand whether these changes play a mechanistic role [443].

### 5.1. P1 Receptors

P1 receptors consist of four distinct adenosine receptor subtypes: the A1, A2A, A2B, and A3, with tissue-specific distribution [444,445,446]. Adenosine receptors are present on endothelial cells, vascular smooth muscle cells, liver adipocytes, and different types of leukocytes.

A_1_R, in adipocytes, is antilipolytic and is implicated in adipogenesis and leptin production [447,448]. Pharmacological stimulation of A_1_R decreased plasma levels of FFAs, glycerol, and triglycerides in Zucker and HFD fed rats. In rats, white adipocytes were more responsive than brown adipocytes tissue to inhibiting lipolysis by activating A_1_R [449]. Adenosine receptors in white and brown adipocytes mediate insulin signaling and age-related changes in adipose tissue [450]. A_1_R KO mice have increased fat mass and body weight and impaired glucose tolerance and insulin sensitivity [451]. Conversely, mice overexpressing the A_1_R in adipose tissue are protected from obesity-induced insulin resistance [452]. A_2B_ adenosine receptor knockout mice fed an HFD developed hallmarks of the metabolic syndrome and T2DM (such as insulin resistance and increased insulin levels and were more obese than wild-type littermates. Stimulation of A_2B_R reversed age-related and obesity-associated sarcopenia and restored skeletal muscle function and mass [453]. Deletion of A_2B_ in skeletal muscle in mice caused sarcopenia, diminished muscle strength, and reduced brown adipose tissue and energy expenditure [453]. A_2B_ adenosine receptor expression in the subcutaneous fat of obese patients is associated with increased BMI and insulin receptor substrate 2 (IRS-2) mRNA expression. The ability of the A_2A_ receptor to regulate BAT thermogenesis and the browning of WAT could increase energy consumption as a treatment for obesity [450].

Adenosine receptors were shown to have an essential role in glucose homeostasis [454]. Several studies have linked adenosine receptor blockade with reversing insulin resistance in skeletal muscle isolated from diabetic animals [455]. NECA, an adenosine receptor agonist, increased β-cell mass, decreased insulin secretion and increased blood glucose levels [456]. Genetic KO of A_1_ receptor increased fasting glucose levels and insulin secretion but decreased insulin sensitivity in muscles and adipose tissue due to decreased glucose uptake [456]. A_2A_R activation stimulates insulin secretion in mouse islets which are reversed by pretreatment with the A2A adenosine receptor antagonist, SCH58261 [457]. A_2B_ receptors on endothelial cells and macrophages are increased in T2D, enhancing the production of IL-6 and stimulating an inflammatory response and insulin resistance in skeletal muscle, adipose tissue, and liver and pancreas [458]. Stimulation of A_2B_ adenosine receptors inhibited adipogenesis and stimulated the differentiation of these cells toward an osteoblastic phenotype. A_2B_^−/−^ adenosine knockout animals fed a standard diet displayed increased adipose tissue inflammation, which was characterized by increased production of proinflammatory cytokines, chemokines, inflammatory macrophage markers and reduced production of IL-10.

Loss of A_2A_R^−/−^ in apoE^−/−^ mice increased plasma cholesterol in the LDL particle and increased intima formation suggesting an anti-atherosclerotic role for the receptor [459]. This contrasts with the observations made in vitro with A_2A_R agonist CGS-21680 in human macrophages and in cultured peritoneal macrophages, where A_2A_R had a pro-atherosclerotic role [459]. A_2B_R is protective against atherosclerosis, and agonists were shown to reduce vascular lesion formation [460]. Endothelial cells lacking the A_2B_R have elevated levels of ICAM-1, P-selectin, and E-selectin [460]. A_2B_R protects platelets from excessive thrombus formation, while A_2B_R KO mice had increased P_2_Y1R expression, an activator of platelet aggregation [461,462]. Vascular smooth muscle cells lacking expression of this receptor have an increased proliferation rate [463].

A_2B_^−/−^ on C57BL/6J background has reduced heart rate when fed HFD. A_1A_R null mice have elevated blood pressure and heart rate at baseline on low sodium diets [464]. In addition, adenosine signaling through the A_2A_ and the A_2B_ provides a potent vasodilatory effect on mean arterial pressure [465,466]. In cardiomyocytes, adenosine increases eNOS activity and protects from mitochondrial damage [467]. A_2A_^−/−^ mice have increased blood pressure and decreased heart rate, which is strain-dependent [468,469]. Thus, targeting the A_2A_R could be a valuable tool for lowering blood pressure. In the vessel, endothelial A_2A_R leads to an increase in nitric oxide production because of activation of the eNOS. A_2A_R confers pulmonary arterial hypertension in mice, while in situ adenosine infusion reduced pulmonary vascular resistance in humans [362,470]. Furthermore, the A_2A_R agonist attenuates the progression of pulmonary hypertension in an experimental model of PAH [471].

In injured tissues, the anti-inflammatory effects of adenosine have a beneficial role [472]. Activation of A_2B_R or A_2A_R in the bone marrow reduces inflammation, increases anti-inflammatory cytokine IL-10, and reduces pro-inflammatory cytokines such as TNF-α [473]. In accordance, A_2B_R KO in macrophages increases TNF-α and IL-6 levels [474]. Similarly, A_3A_R activation in macrophage cell lines inhibits LPS-stimulated cytokine release [475]. Activating adenosine receptors in bone marrow cells in a model of restenosis and angioplasty was shown to be due to its effect on releasing proinflammatory cytokines [476].

Given the functional role of adenosine in many diseases makes it an attractive therapeutic target [476]. However, there are several limitations to developing adenosine therapeutics [477]. Adenosine receptors are widely distributed throughout the body, and their tissue-specific roles are not well understood. Therefore, adenosine can have detrimental or protective effects depending on the nature of the tissue injury and associated pathological conditions. The second drawback is that the opposite effects of adenosine receptor activation at different stages of various disorders make it more complex as a therapeutic target. Thus, it was suggested that focusing on a partial agonist and indirect modulation using allosteric enhancers might maximize the benefits [478].

### 5.2. P2 Purinergic Receptors

P2 receptors are composed of P2XRs, the ligand-gated cation channels, and the P2YR subtypes are stimulated by different endogenous nucleotides [442]. The different receptors are coupled to either Gq to activate PKC, Gs to activate cAMP, and Gi to inhibit cAMP [479,480,481]. Purinergic receptors P_2_Y_1_R, P_2_Y_2_R or P_2_Y_4_R, and P_2_Y_11_R promote adipogenic differentiation of stem cells derived from bone marrow or adipose tissue [447]. Studies in isolated rat white adipose tissue show that activating different P2Rs, activated lipolysis and inhibited insulin-induced leptin production [373]. In contrast, P_2_Y_1_R^−/−^ mice had lower plasma leptin levels on a regular diet but were increased in mice on an HFD. P_2_Y_13_, and P_2_Y_14_, receptors are anti-adipogenic [482,483].

Mouse P_2_Y_4_ receptors are negative regulators of cardiac adipose-derived stem cell differentiation and cardiac fat formation. Stimulation of P_2_Y_4_R by UTP or MRS4062 inhibited adiponectin expression and secretion in cardiac adipocytes. Conversely, P_2_Y_4_R KO mice showed increased adiponectin secretion in hypoxia and was cardioprotective. Rodent BAT expresses several P2Y receptors, and stimulation with ATP leads to exocytosis and heat production [484,485]. In a more recent study, ATPγS enhanced UCP1 expression and induced browning in BAT in conditions of low adaptive thermogenesis and β-adrenergic receptor KO mice where P_2_Y_12_R are overexpressed [486].

P_2_Y_6_R is ubiquitously expressed in many organs and tissues, where it is involved in glucose homeostasis, insulin resistance, obesity, hypertension, and electrolyte homeostasis [487]. P_2_Y_6_ function in obesity was examined using mice with KO of P_2_Y_6_ in either adipose tissue or skeletal muscle. On HFD, it was observed that adipose tissue KO mice gained less weight, while skeletal muscle KO mice gained more weight than controls. The adipose tissue KO mice had improved insulin sensitivity than skeletal muscle KO, which was insulin resistant indicating a tissue-specific effect of P_2_Y_6_ receptors [488]. Neuron-specific inactivation of P_2_Y_6_R reduces food intake and improves systemic insulin sensitivity in obesity. P_2_Y_1_ receptor-mediated insulin stimulating responses of β cells and the pancreas vascular bed were preserved in the STZ-diabetic rat pancreas [489]. P_2_Y_1_ receptors were present in islet capillaries; islets isolated from P2Y_1_^−/−^ mice showed higher insulin secretion in response to glucose than wild-type. Global P_2_Y_13_ deficiency leads to an improved outcome in metabolic syndrome with increased protection against developing insulin resistance, as shown through an improved glucose tolerance and basal glucose levels, a decelerated weight gain despite comparable food consumption, and a better metabolic turnover. P_2_Y_1_ and P_2_Y_13_ mediate β-cell apoptosis.

In the heart, ATP concentration increases in interstitial spaces by ischemia, mechanical stress, increased workload, and contractile agents [490,491]. Pharmacological strategies and the use of transgenic and knock-out animals have revealed differences in the function of P2 receptors in the ‘heart. Mice treated with P_2_Y_2_R-selective agonists and P_2_Y_2_R knockout mice are protected from ischemic damage [491,492,493]. P_2_Y_2_R signaling down-regulates AT1R density in pressure-overloaded hearts and prevents pathological remodeling [494,495,496]. P_2_Y_4_R^−/−^ mice are protected during myocardial infarction by reductions in microvascular hyperpermeability and neutrophil infiltration. They also have smaller hearts, reduced exercise capacity, and adaptive hypertrophy.

In contrast, cardiomyocytes from P_2_Y_6_R-deficient mice are more proliferative than those from wild-type mice, resulting in postnatal hypergrowth of the heart. P_2_Y_2_R and P_2_Y_6_R expression is upregulated in chronic heart failure and dystrophic cardiomyopathy in mice [495,497,498]. The P_2_Y_11_ receptor has a wide distribution in all cell types relevant for cardiovascular pathology: cardiomyocytes, fibroblasts, endothelial and immune cells [495]. P_2_Y_11_R binds ATP, β-NAD, and NAADP, and its stimulation has a protective role in myocardial I/R, inhibits cardiac fibroblast proliferation, and has positive inotropic effects in murine cardiomyocytes. P_2_Y_11_R activation was shown to delay graft rejection by attenuating the local immuno-inflammatory response in an in vivo murine model of heterotopic heart transplantation. Polymorphism of P2Y_11_R is associated with an increased risk of acute myocardial infarction [499].

In the vasculature, extracellular nucleotides participate in local control of blood flow through activation of P2 receptors [500]. P_2_Y_11_R agonist decreased vasoactive factors eNOS and ET-1 in human ECs and prevented proliferation and switch of SMC towards a synthetic phenotype [501]. Vascular smooth muscle cells express P_2_Y_12_ receptors, which mediate contractile function after stimulation with ADP [502]. P_2_Y_6_R is upregulated with aging and under pathological conditions such as arterial inflammation and hemodynamic overload of the heart, and it contributes to the pathogenesis of cardiovascular remodeling in mice [490,503]. Allosteric modulation of P_2_Y_6_R by MRS2578, a P_2_Y_6_R antagonist, suppresses heterodimer formation by P_2_Y_6_R with AT1R, which reduces the risk for Ang II-induced hypertension [504]. These results suggest that age-dependent increases in P2Y_6_R expression determine Ang II’s pathological vascular effects and cardiovascular risk. P_2_Y_6_R-deficient mice exhibit a decrease in Ang II-induced pathological arterial remodeling in response to hypertension than wild-type mice [503] P_2_Y_1_R and P_2_Y_13_R are expressed in pulmonary artery vasa vasorum endothelial cells and functionally involved in intracellular and mitochondrial Ca^2+^ regulation associated with pathologic angiogenic expansion of the vasa vasorum network [479,505].

Studies in knockout mice revealed that P_2_Y_1_R, P_2_Y_2_R, P_2_Y_6_R, P_2_Y_12_R are pro-atherogenic, and P_2_Y_13_R is protective in atherosclerosis [506,507]. Endothelial-specific P_2_Y_2_R deletion prevents atherosclerosis in apolipoprotein E null (ApoE^−/−^) mice [508]. P_2_Y_13_R deficiency exacerbates atherosclerosis in mice. Bone-marrow transplantation assays showed that non-hematopoietic-derived P_2_Y_13_R protects against atherosclerosis development by mediating hepatobiliary reverse cholesterol transport [509].

P2Y receptors regulate immune cell function, including phagocytosis cytokine production and lymphocyte activation [510]. As DAMPs purinergic metabolites control CVD-inflammation. P2Y receptors are present in lymphoid tissues such as the thymus, spleen, and bone marrow, where they are expressed on lymphocytes, macrophages, dendritic cells, neutrophils, eosinophils, mast cells, and platelets [511]. In macrophages, P_2_Y_6_R promotes the secretion of pro-inflammatory cytokines, which are also involved in atherosclerotic lesion development in high fat-fed LDLR^−/−^ mice [512]. Bone-marrow transplantation assays also revealed the importance of non-hematopoietic derived P_2_Y_1_R, P_2_Y_6_R, and P_2_Y_12_R in atherosclerosis. Studies of P_2_Y_1_R- and P_2_Y_12_R-deficient mice revealed the importance of purinergic receptors for platelet aggregation and thrombus development [513].

P_2_Y_2_ and P_2_Y_11_ promote atherosclerotic inflammation and attract inflammatory cells to the atherosclerotic plaque [514]. The P_2_Y_2_ receptors release free radicals in human macrophages [515]. P_2_Y_1_ receptor knockout mice exhibit reduced plaque area occupied by macrophages and the decreased total amount of atherosclerotic lesions in ApoE knockout mice [515].

Consequently, P_2_Y receptors have a role in various infectious, autoimmune, and inflammatory diseases. P_2_Y_12_R ligands are therapeutics, and an additional 209 ongoing clinical trials with agents targeting purinergic signaling are ongoing. These clinical trials test purinergic agents to treat diverse diseases, including cardiovascular diseases caused by or associated with immune dysregulation. In addition, according to https://clinicaltrials.gov/ (accessed on 20 August 2021), there are diseases, metabolic syndrome, diabetes, kidney, and respiratory disease [516].

In summary, it is apparent from the various studies metabolites and their receptors play an essential role in the cross-talk between metabolism and inflammation in maintaining homeostasis. However, the expression and role of multiple GPCRs with apparently similar functions in the same tissue are not understood. However, based on KO studies, each of them contributes to physiology. Furthermore, studies in transgenic and knock-out models suggest that other factors may contribute to the outcomes. Variations in diet and the gut microbiome can result in differences in concentrations in metabolite ligands. Also, the ligand preference of these receptors in vivo is still not clear. Therefore, future studies will have to correlate the gut microbiome and the diet composition with the different metabolites and their receptors in the tissues of interest.

The discovery of key metabolites as ligands for specific GPCRs has significantly broadened our understanding of metabolic signaling and provides several novel potential drug targets. Changes in the expression and function of the receptors highlighted in this review can influence the development and progression of metabolic diseases (Table 1 and Figure 1). However, drug development remains challenging in many cases due to limited or conflicting data, a lack of understanding of basic receptor pharmacology, species-specific effects, tissue-specific effects, and variability in results from different laboratories have hindered the translation of many of these studies into therapeutic compounds. More rigorous early-stage target validation is required, including improved compound screening techniques and novel targeting mechanisms, including signaling bias and allostery, to avoid toxic side effects, especially in cases where tissue-specific effects vary. Several clinical trials are testing candidate ligands in different diseases. We compiled ongoing clinical trials targeting metabolic receptors in Table 2.

Table 1 Summarizes the physiological and pathological roles of metabolite receptors in cardiometabolic diseases and their tissue expression.

Table 2 Lists the ligands for metabolite GPCRs receptors, their IUPHAR #, in clinical trials.

## Figures and Tables

**Figure 1 cells-10-03347-f001:**
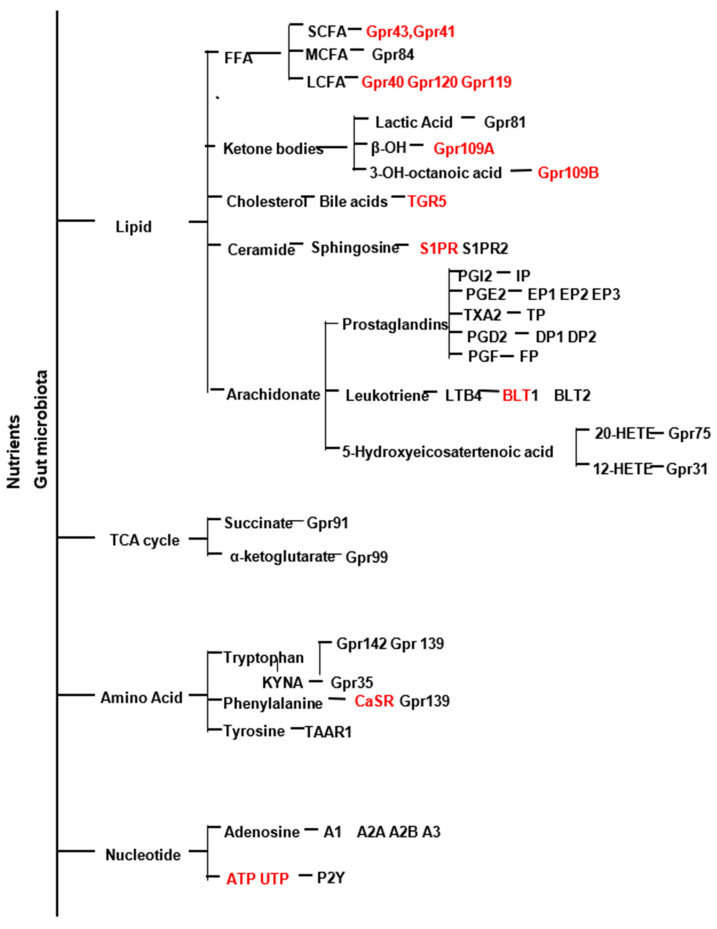
The schematic shows the different groups of metabolites and their ligands. Those marked in red are in clinical trials for metabolic disease.

**Table 1 cells-10-03347-t001:** Metabolites–Receptors–Physiological Actions.

GPCR	Physiological/Pathological Action	Tissue Expression
Short Chain Fatty Acid Receptors		
FFAR2/GPR43	↓ fat lipolysis, ↓ insulin sensitivity, ↑ anorectic hormones, GPR43^−/−^ are mice obese on a regular diet and protected from weight gain on HFD	brain and lung tissues, with lesser expression in heart, skeletal muscle, intestine, liver–kidney adipocyte artery, leukocytes.
FFAR3/GPR41	GPR41^−/−^ ↑ insulin secretion, ↑ cardiac hypertrophy ↑, blood pressure	brain and lung tissues, with lesser expression in heart, skeletal muscle, intestine, liver–kidney adipocyte artery, leukocytes
Olfr78	olfr78 ↑ blood pressure ↑ inflammation	Vascular cells
Medium Chain Fatty Acid Receptors		
GPR84	Pro-inflammatory, ↑ diabetes, atherosclerosis, heart failure, and ↑ fatty acid metabolism obesity. ↑ Fibrosis in lung	immune cells, lung, lymph nodes, and adipose tissue
Long-Chain Fatty Acid Receptors		
FFAR1/GPR40	↑ obesity, ↑ Insulin secretion, adipogenesis. Studies with GPR40^−/−^ mice on fat metabolism controversial may depend on fat and glucose levels suggest a homeostatic role	pancreatic β cells, intestinal cells, adipocytes, and liver, immune cells
FFAR4/GPR120	protective in obesity, blood pressure, atherosclerosis and is anti-inflammatory	pancreatic β cells, intestinal cells, adipocytes, and liver, immune cells
GPR119	GPR119 agonists lowered blood glucose, protective in atherosclerosis, anorectic but lowered metabolism in heart and skeletal muscle	GPR119 also expressed in cardiac and skeletal muscle
Ketone Body Receptors		
HCA1/GPR81	↑ Insulin sensitivity in mouse models of diabetes regulation of renal vascular resistance by modulation of the endothelin. Possible anti-inflammatory	adipocyte; low kidney, skeletal muscle, and liver levels adipocytes and immune cells, heart, vascular
HCA2/GPR109A	↓ fat accumulation, Agonists protective in systemic and pulmonary hypertension ↓ lipolysis and anti-inflammatory	adipocyte; low kidney, skeletal muscle, and liver levels adipocytes and immune cells, heart, vascular
HCA3/GPR109B	GPR109B is expressed only in human’s anti-inflammatory and inhibits adiposity	
Bile Acid Receptors		
TGR5	Bile acid homeostasis, energy homeostasis, insulin signaling, and inflammation. Dysfunction causes cholestatic liver diseases, dyslipidemia, fatty liver diseases, cardiovascular diseases, and diabetes	the small intestine, stomach, liver, lung, placenta, and spleen
Ceramide		
SIP1R	↑ obesity, insulin resistance, hyperglycemia, dyslipidemia, and hypertension. Proinflammatory in macrophages	Macrophages endothelial cells
S1P2R		Pancreatic beta cells, Skeletal muscle
Prostanoids		
Prostaglandins	↓ Protective against obesity-induced inflammation substrate analogs improve insulin sensitivity, protective in diabetic nephropathy, ^−/−^ mice show increased intimal hyperplasia, atherosclerosis, and hypercoagulability and thrombus formation	Vascular, T cells, platelets macrophages, pneumocytes, smooth muscle cells, and fibroblasts vascular cells, platelets, macrophagesleukocytes, including granulocytes, T Cells, dendritic macrophages, and vascular smooth muscle cells
PGI		
TXA2	↑ Obesity, ↑platelet aggregation, modified by insulin sensitivity, inflammatory in macrophages, glucose, insulin resistance, and triglycerides.	
PGE2	↑ Obesity. EP3 receptor inhibitors reversed obesity-induced tissues inflammation. In the kidney vasculature, EP2 and EP4 ↑vasodilation, whereas EP1 and EP3 ↑ vasoconstriction. Cardiomyocyte-specific deletion of the EP4 ↑cardiac dysfunction after myocardial infarction	
PGF	PGF_2_*_α_* suppresses an early phase of adipogenesis. FP^−/−^ mice ↓ blood pressure, coincident with a reduction in plasma renin concentration, angiotensin, and aldosterone	
Leukotrienes	LTB4 antagonists and BLT-1^−/−^ mice are protected from HFD-induced insulin resistance and decrease macrophages and T cells infiltration in adipose tissue. Inhibition of BLT1 is protective in atherosclerosis.	
BLT1		
BLT2		
Hydroxy-eicosatetraenoic acids	agonists promote vascular smooth muscle contraction, endothelial dysfunction, inflammation, and cell proliferation. The 20-HETE antagonist attenuated weight gain and prevented hyperglycemia	T cells, platelets macrophages, pneumocytes, smooth muscle cells, and fibroblasts vascular cells, platelets, macrophages
20-HETE/GPR75	12-HETE increases oxidative stress and modulates inflammation via interaction with GPR31. GPR31^−/−^ mice protect obese HFD fed mice from glucose intolerance and improve insulin secretion in cytokine-treated islets.	leukocytes, including granulocytes, T Cells, dendritic macrophages, and vascular smooth muscle cells
12-HETE/GPR31		
TCA Cycle Metabolites		
GPR91/SUCNR1	Increased BMI, insulin, increase in adipose tissue protects from liver lipo-toxicity. An increase in the liver promotes atherosclerosis, vasorelaxant, increased in Metabolic syndrome, proinflammatory	white adipose tissue, liver, heart intestine, spleen, and immune system cells, including dendritic cells vascular cells
GPR99/*Oxgr1*	decreased concentrations of αKG in the right and left ventricles of mice exposed to hypoxia promote cardiac hypertrophy	brain, lung, kidney, heart, and skeletal muscle
Amino Acid Receptors		
GPR142	increased during fasting and decreased in DIO improved insulin sensitivity delayed the onset and progression of diabetes, and is anti-inflammatory	pancreas and the immune system pancreatic islets and skeletal muscle, with relatively higher levels in adult lung, small intestine, colon, and stomach.
GPR35	lipid metabolism, thermogenic, and anti-inflammatory gene expression in adipose tissue	kidney and parathyroid glands and to a lesser extent in lungs, skin, intestine, b
CasR	CasR vascular tone, metabolic processes in vascular cells, lung and neuronal development, or cardiac function, promote glucose-induced insulin secretion Pro-inflammatory	
GPR139	GPR139^−/−^ mice are lean, target for obesity and T2D	hypothalamus, pituitary, and habenula rain, and vasculature on immune cells
TAAR1	Increases insulin secretion and glucose tolerance decreased food intake and body weight in a diet-in. increase inflammatory cytokine gene expression in non-polarized and LPS-polarized BMDM Mets and inversely correlated with multiple biomarkers of inflammation and cardiometabolic risk factors such as body mass index and blood pressure	TAAR1 neuronal cells and peripheral organs such as the GI tract immune cells
Nucleoside Receptors		
P1R	A1 antilipolytic and is implicated in adipogenesis mediate insulin signaling and age-related changes in adipose tissue. Adenosine ↑ vasodilation and is anti-inflammatory	endothelial cells, vascular smooth muscle cells, liver, adipocytes, and different types of leukocytes.
A_1_R		
A_2A_R	A_2A_^−/−^/ApoE^−/−^ shows an anti-atherosclerotic role for A_2A_. In macrophages, A_2A_ shows a pro atherosclerotic role, is anti-inflammatory in macrophages.	
A_2B_R	A_2B_^−/−^ mice develop metabolic syndrome and T2D ↑ inflammation of adipose tissue. ↑ browning of fat, ↓ atherosclerosis is anti-inflammatory in endothelial cells ↓ platelet aggregation. Anti-inflammatory in macrophages	
P2R	P2 receptors: All P2 receptors are adipogenic.	endothelial cells, vascular smooth muscle cells, liver adipocytes, and different types of leukocytes Heart, platelet, skeletal muscle, neuron, intestine
P_2_Y_1_	P_2_Y_1_↑ β-cell apoptosis. P2Y1^−/−^ ↑ insulin secretion. ischemic diseases, pressure overload hypertrophy, and ↓myocardial infarction in the heart	
P_2_Y_2_	Pro-atherogenic, proinflammatory	
P_2_Y_4_	promote thermogenesis in BAT	Heart, lung, skeletal muscle, spleen, kidney lung, vascular smooth muscle, brain, Liver
P_2_Y_6_	Insulin resistance, obesity, hypertension, and electrolyte homeostasis. P_2_Y_6_^−/−^ in adipose tissue is insulin-sensitizing, whereas skeletal muscle KO is insulin resistant	
P_2_Y_11_	P_2_Y_11_ is protective in ischemia, promotes hypertension, pro and anti-inflammatory roles reported, ↓ insulin-stimulated leptin in adipocytes and ↑ lipolysis	Spleen, intestine, Immune cells Heart, lung, spleen, intestine, brain
P_2_Y_12_	Pro-atherogenic	
P_2_Y_13_	P_2_Y_13_^−/−^ mice showed improved metabolic parameters	

**Table 2 cells-10-03347-t002:** Metabolites–Receptors–Clinical Trials.

Natural Metabolite-CAS#	Receptor	Trial Name	Drug	Clinical Trials Relevant to This Review
2-Propionate-butyrate (CAS156-54-7)	GPR43/GPR41 (FFA2 and FFA3)	The Effect of PPI Therapy on Weight, Gut Microbiome, and Expression of GPR41 and GPR43	Diet questionnaire	NCT02457104
	GPR43 (IUPHAR (#226))	Effect of a Synbiotic on the Gut Microbiota and Adiposity-related Markers in Healthy Overweight Subjects	Dietary Supplement Synbiotic	NCT0215182
	GPR41 (IUPHAR # 227) gene#2865)	Effect of fermentable carbohydrates in glucose homeostasis	Dietary supplement Inulin and cellulose	NCT01841073
9-a-linoleic acid (CAS#60-33-3)	GPR40 (FFA1) (IUPHAR-# 311) (gene#2864)	Determination of Safety, Tolerability, Pharmacokinetics, Food Effect and pharmacodynamics of Single and Multiple Doses of P11187	GPR-40-agonist P11187	NCT01874366
				NCT04703361
		Effect of Dietary oils as GPCR agonists on Glucose tolerance	Dietary Supplement Pine nut and olive oil	NCT03774095
	GPR109A (HCA2) (IUPHAR-# 312) (gene #338442)	Short-term Effect of Extended-release Niacin on Endothelial Function.	Niacin Na-OHB	NCT01942291;
8-palmitate, PA (CAS#2210-62-0)	GPR120 FFA4 (IUPHAR-# 127) (gene#338557)	Expression of G-protein Coupled Receptor 120 (GPR120) Receptor in Adipose Tissue of German Diabetes Center (GDC) Cohort Subjects	Association of R270H mutant with T2D and expression of GPR120 in T2D	NCT03285750
10-oleoylglycerol (CAS#111-03-5)	GPR119 (IUPHAR-# 126) (gene#139760)	-Oleoyl Glycerol is a GPR119 Agonist and Signals GLP-1 Release in Humans.	Dietary Supplement: GPR 119 agonist, 2-oleyl glycerol Dietary Supplement: Oleic acid	NCT01043445
		Diet oil supplementation induced release of GLP-1	Diet oil supplementation of olive and carrot oil	NCT01453842
11-Lithocholic acid (434-13-9)	GPBAR1 (TGR5) (IUPHAR #37) (gene#151306)	Effect of Bile Acids on the Secretion of Satiation Peptides in Humans	bile acid (CDCA, chenodeoxycholic acid) oleanolic acid Dietary Supplement: oleic acid	NCT01674946
		The Impact of Gall Bladder Emptying and Bile Acids on the Human GLP-1-secretion	Acetaminophen Metformin Colesevelam CCK-8	NCT01656057
		Importance of Meal Fat Content and Gall Bladder Emptying for Postprandial GLP-1 Secretion in Type 2 Diabetes Patients	GLP-1 secretion via TGR5	NCT01374594
		Bile Acid-induced GLP-secretion. A Study in Cholecystectomized Subjects	GLP-1 secretion via TGR5 in healthy and Cholecystectomized Subjects	NCT01251510
		Effect of Bile Acids on GLP-1 Secretion	Chenodeoxylic acid Colesevelam	NCT01666223
20-HETE	GPR75 (IUPHAR #115) Gene#10936	Genetic and Dietary Predictors of Anti-platelet Response	ASPIRIN	Not recruiting yet
Sphingosine-1-phosphate, (CAS#26993-30-6)	SIPR1(EDG1) (gene#1901) (IUPHAR#275)	Efficacy, Safety and Tolerability of BAF312 Compared to Placebo in Patients With Intracerebral Hemorrhage (ICH).	BAF312	NCT03338998
LTB4	BLT1 IUPHAR#271)	Body Weight, Aspirin Dose and Pro-resolving Mediators	ASPIRIN	Not recruiting yet
PGI2-prostacyclin, (CAS # 63859-31-4)	IP (IUPHAR#345) (Gene#5739)	Evaluation of a New Thermostable Formulation of FLOLAN in Japanese Subjects	FLOLAN	NCT02705807
		Drug Use Investigation for FLOLAN (Epoprostenol) Injection 0.5mg·1.5mg	FLOLAN	NCT01387191
		Inhaled Nitric Oxide and Inhaled Prostacyclin After Cardiac Surgery for Heart Transplant or LVAD Placement	Prostacyclin FLOLAN	NCT01717209
		Epoprostenol in Pulmonary Embolism	Epoprostenol	NCT01014156
Thromboxane TxA2	TXA2-R (Gene#6915) (IUPHAR#346)	Thromboxane Receptor Antagonist to Improve Endothelial Cell Function	ifetroban	NCT03962855
Phenylalanine, and other amino acids.	CaSR (IUPHAR-# 54) (gene#846)	Lipid and Glycogen Metabolism in Patients With Impaired Glucose Tolerance and Calcium Sensing Receptor Mutations	Meal Tolerance Test: Hyperglycemic-hyper insulinemic clamp	NCT02023489
ATP	P2Y12 Gai (IUPHAR#328) (gene#64805)	Aspirin Impact on Platelet Reactivity in Acute Coronary Syndrome Patients on Novel P2Y12 Inhibitors Therapy	Aspirin P2Y12 inhibitors	NCT03190005
		Comparison Between P2Y12 Antagonist Monotherapy and Dual Antiplatelet Therapy After DES	Aspirin P2Y12 inhibitors combination	NCT02079194
		Association Between Genetic Variant Scores and P2Y12 Inhibitor Effects	P2Y12 inhibitors	NCT04580602
		Aspirin Impact on Platelet Reactivity in Acute Coronary Syndrome Patients on Novel P2Y12 Inhibitors Therapy	P2Y12 inhibitors	NCT02049762
		Tailoring P2Y12 Inhibiting Therapy in Patients Requiring Oral Anticoagulation After PCI	Ticagrelor Clopidogrel	NCT04483583
		ComparisoN of ticAgrelor vs. Clopidogrel in endoTHeliAl Function of COPD patieNts	Ticagrelor Clopidogrel Aspirin	NCT02519608

## Data Availability

Not applicable.

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
