# Peer review of "Metabolite G-Protein Coupled Receptors in Cardio-Metabolic Diseases"

_cells, 2021, doi:10.3390/cells10123347_

Round 1
Reviewer 1 Report
This is a very comprehensive review on metabolite GPCRs and their relationship to cardio-metabolic disease. It is especially good on the likely functions of the receptors. There is rather little about the detailed pharmacology of the receptors, particularly selective agonists and antagonists, but since that is available from other sources such as the IUPHAR Guide to the Receptors, the authors are probably wise to exclude this. I would however suggest that they cite this (or an equivalent) source; I cannot see it in the references. As a non-specialist in the field, whilst I was very impressed with the breadth and depth of the review, at times I was overwhelmed by the information. The authors might consider a few summary paragraphs at the end of each of the major sections, to give an overview of what they have just discussed. Another option might be more tables or some figures to achieve the same end.
Author Response
Reviewer 1
We thank the reviewer for critiquing the review and their suggestions to improve the presentation. We have addressed their concerns in the main text with track changes. In addition, we have included a Table and a new graphical abstract to clarify the role of the receptors in the tissues.
There is rather little about the detailed pharmacology of the receptors, particularly selective agonists and antagonists, but since that is available from other sources such as the IUPHAR Guide to the Receptors, the authors are probably wise to exclude this.
We have listed the CAS# for the agonists, the IUPHAR # for the receptors in the clinical trials in Table 2. We have also listed the IUPHAR guide reference in the legends for the table.
The authors might consider a few summary paragraphs at the end of each of the major sections to overview what they have just discussed. Another option might be more tables or some figures to achieve the same end.
We have included a summary statement for the receptors. We have also added a new Table 1 where we summarized the functions and tissue expression for the GPCRs.
Reviewer 2 Report
This review article on metabolite-activated GPCRs is extremely comprehensive with a very large number of bibliographic references.
It illustrates well the role of this class of receptors in the regulation of food intake, lipid and carbohydrate metabolism, vascular physiology and inflammation.
While the current organization of this very long article, metabolite by metabolite then receptor by receptor is certainly useful for finding specific information, it makes the whole reading rather tedious.
Several of the described receptors are expressed in the same tissues, have comparable signaling mechanisms and redundant effects. As the experimental methods that have been used to study these GPCRs are also quite variable, it is difficult for the reader to get an idea of ​​the relative importance of a particular activation pathway and receptor in relation to other receptors and other pathways in the same tissue and for the same physiological or pathological context.
One of the ways to improve the article would be, for example, to add summary diagrams grouping tissue by tissue the main receptors and ligands as well as the main functional outputs
Take-home messages at the end of the article would also be welcome
A list of all the abbreviations used in the article seems essential to me
Minor points
some phrases must be reviewed because they are difficult to understand (ex: lines 83-85; 632-634)
Some abbreviations are not explained the first time they are used
Author Response
Reviewer 2
We thank the reviewer for critiquing the review and their suggestions to improve the presentation. We have addressed their concerns in the main text with track changes. In addition, we have included a Table and a new graphical abstract to clarify the role of the receptors in the tissues
Several of the described receptors are expressed in the same tissues, have comparable signaling mechanisms and redundant effects. As the experimental methods that have been used to study these GPCRs are also quite variable, it is difficult for the reader to get an idea of ​​the relative importance of a particular activation pathway and receptor in relation to other receptors and other pathways in the same tissue and for the same physiological or pathological context.
We agree that groups of receptors have a similar role and are expressed in each tissue. We predict that the availability of the ligand for the receptors may determine which receptor is utilized. We think that ligand availability will depend on dietary intake, the gut microbiome, and metabolizing enzyme status. This can vary in individuals significantly and modify levels of metabolites and outcomes. Since the detailed signaling pathways for each ligand are understudied, the nuances in outcomes by activation of the different receptors in the tissue may be different. Whether some have a physiological role and others are expressed in pathological conditions is also not known, emphasizing the need for more studies,
One of the ways to improve the article would be, for example, to add summary diagrams grouping tissue by tissue the main receptors and ligands as well as the main functional outputs.
We agree with the reviewer that a figure would help visualize the different receptors in a given tissue. Therefore, we have included a graphical abstract figure of the receptors expressed in the different metabolic tissues. We have also included a summary table (Table 1) with the functions of the receptors. We hope that this addresses the concerns of the reviewer. We have included the list of abbreviations used. We have summarized wherever possible the take-home message for each class of receptors.
Round 2
Reviewer 2 Report
The manuscript has been improved